# Evaluating comparative effectiveness of psychosocial interventions adjunctive to opioid agonist therapy for opioid use disorder: A systematic review with network meta-analyses

**Danielle Rice**[1,2☯], **Kimberly Corace**[1,3,4,5☯], **Dianna Wolfe**[1], **Leila Esmaeilisaraji**[1], **Alan Michaud**[1], **Alicia Grima**[1], **Bradley Austin**[1], **Reuben Douma**[1], **Pauline Barbeau**[1], **Claire Butler**[1], **Melanie Willows**[3,4,5,6], **Patricia A. Poulin**[1,7,8], **Beth A. Sproule**[9,10], **Amy Porath**[11], **Gary Garber**[1,12,13], **Sheena Taha**[11], **Gord Garner**[14], **Becky Skidmore**[1], **David Moher**[1,12], **Kednapa Thavorn**[1,12], **Brian Hutton**[1,12]*

1 Ottawa Hospital Research Institute, Ottawa, Ontario, Canada, 2 Department of Psychology, McGill University, Montreal, Quebec, Canada, 3 Substance Use and Concurrent Disorders Program, The Royal Ottawa Mental Health Centre, Ottawa, Ontario Canada, 4 Department of Psychiatry, Faculty of Medicine, University of Ottawa, Ottawa, Ontario, Canada, 5 Institute of Mental Health Research, University of Ottawa, Ottawa, Ontario, Canada, 6 Faculty of Medicine, Department of Family Medicine, University of Ottawa, Ottawa, Ontario, Canada, 7 Department of Anesthesiology and Pain Medicine, Faculty of Medicine, University of Ottawa, Ottawa, Canada, 8 Pain Clinic, Ottawa Hospital, Ottawa, Ontario, Canada, 9 Department of Pharmacy, Centre for Addiction and Mental Health, Toronto, Ontario, Canada, 10 Leslie Dan Faculty of Pharmacy and Department of Psychiatry, University of Toronto, Toronto, Ontario, Canada, 11 Canadian Center on Substance Use and Addiction, Ottawa, Ontario, Canada, 12 School of Epidemiology and Public Health, University of Ottawa, Ottawa, Ontario, Canada, 13 Public Health Ontario, Toronto, Ontario, Canada, 14 The Community Addictions Peer Support Association (CAPSA), Ottawa, Ontario, Canada

☯ These authors contributed equally to this work.
* bhutton@ohri.ca

## Abstract

### Background

Guidelines recommend that individuals with opioid use disorder (OUD) receive pharmacological and psychosocial interventions; however, the most appropriate psychosocial intervention is not known. In collaboration with people with lived experience, clinicians, and policy makers, we sought to assess the relative benefits of psychosocial interventions as an adjunct to opioid agonist therapy (OAT) among persons with OUD.

### Methods

A review protocol was registered a priori (CRD42018090761), and a comprehensive search for randomized controlled trials (RCT) was conducted from database inception to June 2020 in MEDLINE, Embase, PsycINFO and the Cochrane Central Register of Controlled Trials. Established methods for study selection and data extraction were used. Primary outcomes were treatment retention and opioid use (measured by urinalysis for opioid use and opioid

**Data Availability Statement:** All relevant data are within the paper and its Supporting information files.

**Funding:** Funding for this work was provided by the Canadian Institutes of Health Research (CIHR) grant number 397976. DBR is supported by a Vanier Canada Graduate CIHR Scholarship and received a student grant from the Psychology Foundation of Canada. The funders had no role in the study design, data collection and analysis, decision to publish, or preparation of the manuscript. URL of funder websites: https://cihr-irsc.gc.ca/e/193.html https://vanier.gc.ca/en/home-accueil.html https://psychologyfoundation.org/.

**Competing interests:** BH has previously received honoraria from Eversana (previously Cornerstone Research Group) for the provision of methodologic advice related to systematic reviews and meta-analysis. This does not alter the adherence to PLOS Medicine policies on sharing data and materials. All other authors have no conflicts of interest to disclose. This does not alter our adherence to PLOS ONE policies on sharing data and materials.

abstinence outcomes). Odds ratios were estimated using network meta-analyses (NMA) as appropriate based on available evidence, and in remaining cases alternative approaches to synthesis were used.

## Results

Seventy-two RCTs met the inclusion criteria. Risk of bias evaluations commonly identified study limitations and poor reporting with regard to methods used for allocation concealment and selective outcome reporting. Due to inconsistency in reporting of outcome measures, only 48 RCTs (20 unique interventions, 5,404 participants) were included for NMA of treatment retention, where statistically significant differences were found when psychosocial interventions were used as an adjunct to OAT as compared to OAT-only. The addition of rewards-based interventions such as contingency management (alone or with community reinforcement approach) to OAT was superior to OAT-only. Few statistically significant differences between psychosocial interventions were identified among any other pairwise comparisons. Heterogeneity in reporting formats precluded an NMA for opioid use. A structured synthesis was undertaken for the remaining outcomes which included opioid use (n = 18 studies) and opioid abstinence (n = 35 studies), where the majority of studies found no significant difference between OAT plus psychosocial interventions as compared to OAT-only.

## Conclusions

This systematic review offers a comprehensive synthesis of the available evidence and the limitations of current trials of psychosocial interventions applied as an adjunct to OAT for OUD. Clinicians and health services may wish to consider integrating contingency management in addition to OAT for OUD in their settings to improve treatment retention. Aside from treatment retention, few differences were consistently found between psychosocial interventions adjunctive to OAT and OAT-only. There is a need for high-quality RCTs to establish more definitive conclusions.

## Trial registration

PROSPERO registration CRD42018090761.

## Introduction

In recent years, the illicit use of opioids has risen at alarming rates [1–3], which has contributed to substance use disorder, overdose, and increasing rates of opioid-related death [4]. COVID-19 has exacerbated this public health crisis with increasing numbers of overdoses and fatalities occurring within North America [5, 6]. Between 2016 and 2019, more than 15,000 Canadians died from apparent opioid use [7], with 78% of accidental opioid-related deaths involving fentanyl and fentanyl analogues [8]. In 2018, 67,367 deaths within the United States were attributed to an overdose involving opioids [9]. Problematic opioid use has also been prevalent in Europe, where more than 80% of drug-related deaths in 2017 were related to opioid use [10]. Similar issues exist in Asia, where two thirds of all individuals using opioids have been described as engaging in problematic opioid use [11]. Both the non-medical use of prescription opioids as well as the use of illicit opioids have contributed to the opioid crisis. An

increase in the contamination of illicit drugs with fentanyl in North America and Europe has been one contributing factor to the increasing rate of overdoses, hospitalizations and mortality from opioid use [12, 13]. These trends have prompted international actions including the development of new guidelines [14], such as the 2017 Canadian Guideline for Opioids for Chronic Non-Cancer Pain [15, 16].

Individuals with an opioid use disorder (OUD) have been found to experience increased healthcare utilization, morbidity, and mortality as compared to individuals without an OUD [17, 18]. Although individuals with any substance use disorder are at an increased risk of harm, injecting opioids has additional health risks such as the transmission of blood-borne viruses, including human immunodeficiency virus (HIV) and hepatitis C virus (HCV) [19].

There are also substantial societal implications of OUD. In Canada, over $135 million CAD was spent in 2015 on the medical management of OUD (i.e., methadone, buprenorphine/naloxone) [20], while the total economic impact of opioid overdose, addiction, and dependence from 2001 to 2017 was estimated as exceeding $1 trillion USD in the United States [21]. Given the significant implications of OUD on well-being, healthcare utilization and societal costs, the identification of effective management strategies is essential.

Clinical guidelines recommend opioid agonist therapy (OAT) as the first-line treatment for OUD, with psychosocial (e.g., motivational interviewing) therapy being routinely offerred [16, 22, 23]. A variety of psychosocial approaches have been used to aid in OUD management, including (for example) cognitive behavioral therapy (CBT), contingency management, and supportive counselling [24, 25]. There is limited research, however, that addresses the efficacy of psychosocial interventions used in conjunction with OAT. Despite a lack of quantitative comparisons of the efficacy of various psychosocial therapies, these treatments are considered by many clinicians and patients to be a vital element of OAT treatment, given the key role that psychotherapy can have in improving treatment retention [16, 22, 26]. Previous systematic and narrative reviews [24, 25, 27] have studied the use of psychosocial interventions delivered in combination with OAT, where the overall efficacy of providing psychosocial interventions was supported; however, these reviews have not conducted quantitative comparisons of effects between psychosocial strategies.

Given the absence of meta-analyses comparing the efficacy of competing psychosocial interventions used with OAT for individuals with OUD, the most appropriate psychosocial therapy to apply as an adjunct to OAT is unknown. Network meta-analysis (NMA) allows for the comparison of multiple therapies in a unified analysis using relevant direct and indirect data [28–30]. NMA is commonly used for evidence synthesis to address research questions that involve comparisons between multiple interventions. Such an analysis would be informative for decision-making to consider the most efficacious psychosocial therapies for treating the rising rate of OUD. Therefore, the objective of this study was to conduct a systematic review incorporating NMA to compare the relative benefits and harms of psychosocial therapies among people with OUD receiving OAT.

## Methods

A protocol was published a priori [31] and registered in PROSPERO (CRD42018090761). The reporting of this review adheres to guidance from the PRISMA Statement for Network Meta-Analyses (see S1 Text) [32].

### Search strategy to identify relevant studies

Searches to identify relevant studies for this review were developed and tested by an experienced medical information specialist (B. Skidmore) in consultation with the review team. The

MEDLINE search strategies were peer reviewed by another senior information specialist using the PRESS Checklist [33] prior to execution. Using the OVID platform, we searched Ovid MEDLINE® ALL, PsycINFO, and Embase Classic + Embase. We also searched the Cochrane Library on Wiley. The study searches were conducted on June 24, 2020. Strategies utilized a combination of controlled vocabulary (e.g., "Opiate Substitution Treatment", "Opioid-Related Disorders/dt [drug therapy]", "Buprenorphine/tu [therapeutic use]") and keywords (e.g., "opioid maintenance", "methadone substitution", "OAT"). Vocabulary and syntax were adjusted by database. Randomized controlled trial filters were used where applicable. Conference abstracts prior to 2016 were removed from Embase and CENTRAL and dissertation abstracts were removed from PsycINFO. S2 Text provides the full search strategies that were used. Reference lists of relevant systematic reviews and the set of included studies were searched for additional studies and were integrated into a PRISMA flow diagram.

### Study eligibility criteria

**Population.**   The review included individuals with problematic opioid use that were receiving OAT, including those with OUD as defined by the Diagnostic and Statistical Manual of Mental Disorders (DSM-5) or diagnosed with opioid dependence as defined by the International Classification of Disease (ICD). Earlier diagnoses such as those defined by the DSM-IV were also eligible for inclusion (i.e., opioid dependence, opioid abuse). No restrictions were put in place regarding age or specialty populations (e.g., pregnant women, or incarcerated individuals).

**Interventions and comparators of interest.**   Psychosocial interventions delivered with OAT (e.g., methadone, slow-release oral morphine, injectable OAT) were of interest. Studies had to include at least one arm with an eligible psychosocial intervention. Studies using control groups of either OAT-only or 'standard medical management' were eligible, as they were expected sources of indirect evidence [28] for NMAs. For inclusion, psychosocial interventions were required to target opioid use (e.g., a study of contingency management that provided rewards for decreased cocaine use rather than opioid use was not considered to be eligible). Studies that included the same psychosocial intervention in each group were excluded, as were studies where only the intensity of interventions, setting, or mode of delivery (e.g., online as compared to in person) differed between groups. Studies that did not include at least two arms receiving the same pharmacological interventions were excluded, given best practice guidelines which include OAT as first line treatment for OUD [16, 23].

A primary list of psychosocial interventions with descriptions was developed a priori (see S3 Text) and included interventions such as contingency management (CM), community reinforcement approach (CRA), cognitive behaviour therapy (CBT), counselling, acceptance and commitment therapy (ACT) and motivational interviewing, amongst other therapy types. Studies that applied inconsistent OATs between groups were excluded given the inability to determine the specific component that could have impacted change (i.e., psychotherapy or pharmacotherapy). In some studies, more than two groups were randomized to interventions but only two interventions were eligible. In these instances, all intervention information was extracted; however, only the eligible arms were included for data analyses (e.g., if participants were randomized to four groups but only two of these groups involved participants receiving OAT). If studies included more than two groups with different pharmacological interventions (e.g., two groups randomized to methadone and psychosocial interventions and two groups with buprenorphine and psychosocial interventions), we included we included only two study arms that applied the same pharmacological intervention based on the OAT that was most frequently reported across all studies. Similarly, if studies included multiple arms with varying

prizes in CM (e.g., two groups received vouchers and two groups received take-home medication), we included the study arms that used the prize reported most often across all studies. Any studies that involved tapering individuals off OAT were also excluded.

**Outcomes.**   The co-primary outcomes of interest were treatment retention at last study timepoint and opioid use. Treatment retention could be reported as a continuous or dichotomous measure based on the individuals continuing to receive treatment in the study. Opioid use, based on urinalysis, could be reported as either abstinence from opioids or opioid use. Thresholds for opioid abstinence and opioid use varied between and within studies and were not consistently dichotomous variables. For example, some studies reported opioid abstinence as a proportion of participants that used opioids less than a specific number of times over a set of weeks. As such, opioid use and abstinence were captured separately and based on the description that each study reported.

Secondary endpoints of interest included self-reported opioid use, abstinence from illicit drug use (including but not limited to cocaine, cannabis, benzodiazepines), alcohol use, dropouts from the psychosocial therapy portion of study (but remaining on OAT), adherence to OAT, HIV/HCV risk behaviours, mental health symptoms (e.g., depression, anxiety), measures of craving, quality of life, and adverse events (e.g., increases in substance use). Outcomes had to be reported separately for at least two eligible study groups to be included (e.g., outcomes reported for all groups combined in the study that were not presented separately by group were not extracted). We sought quantitative data from all reporting formats for the outcomes considered (e.g., mean and standard deviation, frequency, p-values). For studies that reported outcomes in multiple formats (e.g., total abstinence from opioids in weeks and abstinence for more than three weeks), we prioritized presenting the reporting format that was more consistently available across the set of included studies.

**Study designs.**   Only RCTs were included because they would best assess the relative effectiveness of psychosocial interventions, while reducing confounding inherent in other study designs. All other types of studies, including observational studies, case-control studies, case series and case reports, were excluded. Systematic reviews were reviewed to inspect reference lists for additional eligible RCTs, but were not eligible for inclusion. Inclusion was limited to studies published in English or French.

## Screening for eligible studies

Citations identified from the literature searches were imported into DistillerSR Software (Evidence Partners, Inc; Ottawa, Canada). Citations were screened independently by two reviewers based on title and abstract (level 1 screening), and subsequently full-text articles (level 2 screening). Level 1 screening was performed using a liberal accelerated approach (i.e., only one reviewer needed to include a citation, while two reviewers were needed to exclude) [34]. Level 1 citations deemed potentially relevant or lacking sufficient information to exclude were reviewed at Level 2, which was performed by two reviewers independently and in duplicate. Disagreements during full-text screening were resolved by discussion or consultation with a third reviewer (KC) if necessary. Prior to conducting screening at level 1 and level 2, 100 title/abstracts and 15 full texts were piloted by the review team to establish agreement and consistency among reviewers regarding the application of eligibility criteria.

## Process of data collection

Primary data collection from the included studies was performed by two reviewers using a standardized electronic data collection form in DistillerSR. Data were independently extracted by a single reviewer and verified by a second reviewer. Data gathered from the included studies

incorporated information regarding study characteristics (authors, year of publication, journal, countries of data collection, source of funding), participant characteristics (eligibility criteria, number of individuals per group), basic participant demographics (age, sex, race), type of opioid use (prescription and/or illicit), cited rationale for opioid use (e.g., chronic pain), duration of opioid use, mode of use (intravenous versus oral), comorbidities or other unique demographic traits, interventions (names, description, including numbers and duration of sessions, setting and therapist expertise, if described), treatment setting (e.g., community, physician office, penitentiary), and outcomes reported. Type of journal models were also extracted to identify journals that were open access. All intervention names and content were reviewed by a PhD candidate in clinical psychology (DR) in consultation with a clinical expert (KC), when necessary to determine if specific arms of an included study were eligible. Interventions were reviewed and labelled based on their core components to ensure that similar interventions were being combined in quantitative analyses. Reported outcomes were extracted in all formats for all arms of a study to determine the most consistently reported format for each outcome. Study traits were summarized in tabular form to facilitate inspection and discussions with team members regarding study heterogeneity and grouping of interventions. If studies reported on the same cohort (e.g., updates of different follow-up durations), the most complete and up-to-date study information was retained.

## Risk of bias assessments of included studies

Risk of bias (RoB) was evaluated for all studies using the Cochrane RoB tool [35]. The Cochrane RoB tool evaluates seven domains (i.e., random sequence generation, allocation concealment, blinding of participant and personnel, blinding of outcome assessment, incomplete outcome data, selective outcome reporting and "other sources of bias") [35]. Random sequence generation, allocation concealment, and "other sources of bias" were assessed at the study level, while blinding of participants and personnel, blinding of outcomes assessment, incomplete outcome data, and selective outcome reporting were assessed at the level of outcome. Four outcomes were selected by the research team as the "critical outcomes" to be assessed separately; these included treatment retention, opioid use, adherence to OAT and adverse events. The domain for incomplete outcome data was not considered for treatment retention given the overlapping concept of treatment retention and dropout, an approach that has been previously applied for Cochrane reviews of OUD trials [24]. The RoB for blinding of participants and personnel was considered to be high for all studies due to the inherent difficulties in blinding when delivering psychosocial interventions [36]. RoB assessments were conducted independently by two reviewers and disagreements were resolved through discussion or by a third reviewer. Results from RoB appraisals were summarized and reported on an item-by-item basis. RoB was assessed based on the details published in the study, the associated supplementary materials, and available on trial registration websites.

## Approach to evidence synthesis and sensitivity analyses

We planned a priori to undertake NMAs of available direct and indirect evidence using a Bayesian framework for outcomes with sufficient data for analysis in cases where well-connected evidence networks existed, and the transitivity assumption was judged appropriate. Opportunities for such analysis (as well as pairwise meta-analysis) were very greatly limited due to considerable variability in the outcomes measured across trials (leading to disconnected networks of evidence for most outcomes) and the presence of few studies for most treatment comparisons. A descriptive approach to synthesis was thus necessary for most outcomes, though NMA was feasible for one outcome measure (treatment retention). For brevity, we

refer readers to S4 Text for details as to how NMA modeling was performed, including details regarding specifications, assessment of model convergence, estimation of secondary measures of effect, and software considerations. Briefly, random effects NMA was conducted in a Bayesian framework using WinBUGS Software (WinBUGS version 1.4.3, Imperial College and Medical Research Council (MRC) Biostatistics Unit, UK) and R Software (R version 3.5.2, The R Foundation for Statistical Computing). To assess potential for publication bias, comparison-adjusted funnel plots (i.e., plots of the effect estimate from each study against its effect estimate standard error) were generated in Stata (Stata/SE version 15.1, StataCorp LLC, College Station, TX) for studies included in NMAs to assess the potential bias related to the size of the trials, which could indicate possible publication bias [37]. Treatments were ordered by intensity, based on the number of therapy components delivered in addition to OAT.

To assess whether findings from analyses were sensitive to between-study differences in characteristics (e.g., related to enrolled populations or study methods), sensitivity analyses, including subgroup analyses and network meta-regression were planned where sufficient data were available. Unfortunately, the feasibility of sensitivity analyses was also low due to the variable outcome reporting formats and differences in reporting of study characteristics. Again for brevity, we present our a priori plans for secondary analyses in S4 Text, along with the results of secondary analyses that were possible for the treatment retention outcome. Briefly, for this outcome we were able to explore the effects of age, study duration and control group event rate (as a proxy to consider between-study differences in multiple confounders) using meta-regression, as well as publication in potential predatory journals through exclusion from analysis (see S5 Text for a listing of protocol deviations).

### Additional approaches to data synthesis

A structured descriptive synthesis [38] was taken for several endpoints where data were not amenable to meta-analysis. As vote counting approaches are not recommended, summaries of findings based on intervention groups are provided below, while detailed study-by-study data have also been organized and presented in supplements; the latter information follow supplements which detail a map of outcomes reported by each study (S6 Text) and findings from risk of bias appraisals (S7 Text), and can be found within S8 through S24 Texts. Findings reported include the intervention and control group labels; a description of the outcome measures, grouped by similar descriptions between studies; aggregate data reported in each study; follow-up time point (reported in descending order of duration); and author conclusions.

### Reporting of review findings

Both graphical and numerical displays of findings are presented for outcomes, where appropriate. For the NMA performed, a network diagram was generated to display the availability of evidence for the included treatment comparisons; forest plots and league tables were also generated to present its findings. Due to the high volume of outcomes assessed to maximize the value of this review, findings in the review's main text are focused upon the co-primary outcomes of treatment retention and opioid use measured by urinalysis, and appendices have been used to provide details for the remaining outcomes.

## Results

### Extent of literature identified

The literature search identified 17,755 unique citations across databases, and 184 unique citations were identified from hand searching of relevant reference lists. At the level of title/

abstract review, 813 abstracts were judged to be potentially relevant and their full texts were acquired. During full-text review, 72 trials (see S25 Text) met eligibility criteria and were retained for inclusion in the review (see Fig 1) [39–110]. A summary of studies excluded during full-text review, with reasons for exclusion, is provided in S26 Text. Table 1 provides a study-by-study account of additional information including population and key demographics and S1 Data provides detailed accounts of the study accounts of the intervention and comparator groups.

## Study characteristics

The included trials were published between 1978 and 2020 (median year 2009; mode year 2013). The median study sample size was 105 participants (range 14 to 1,015). The majority of trials were conducted in the United States (n = 53, 73.6%) [39, 44, 46, 47, 49, 51–58, 63–71, 73–80, 82–90, 92–95, 97–104, 108, 110], while the remainder were conducted in China (n = 7,

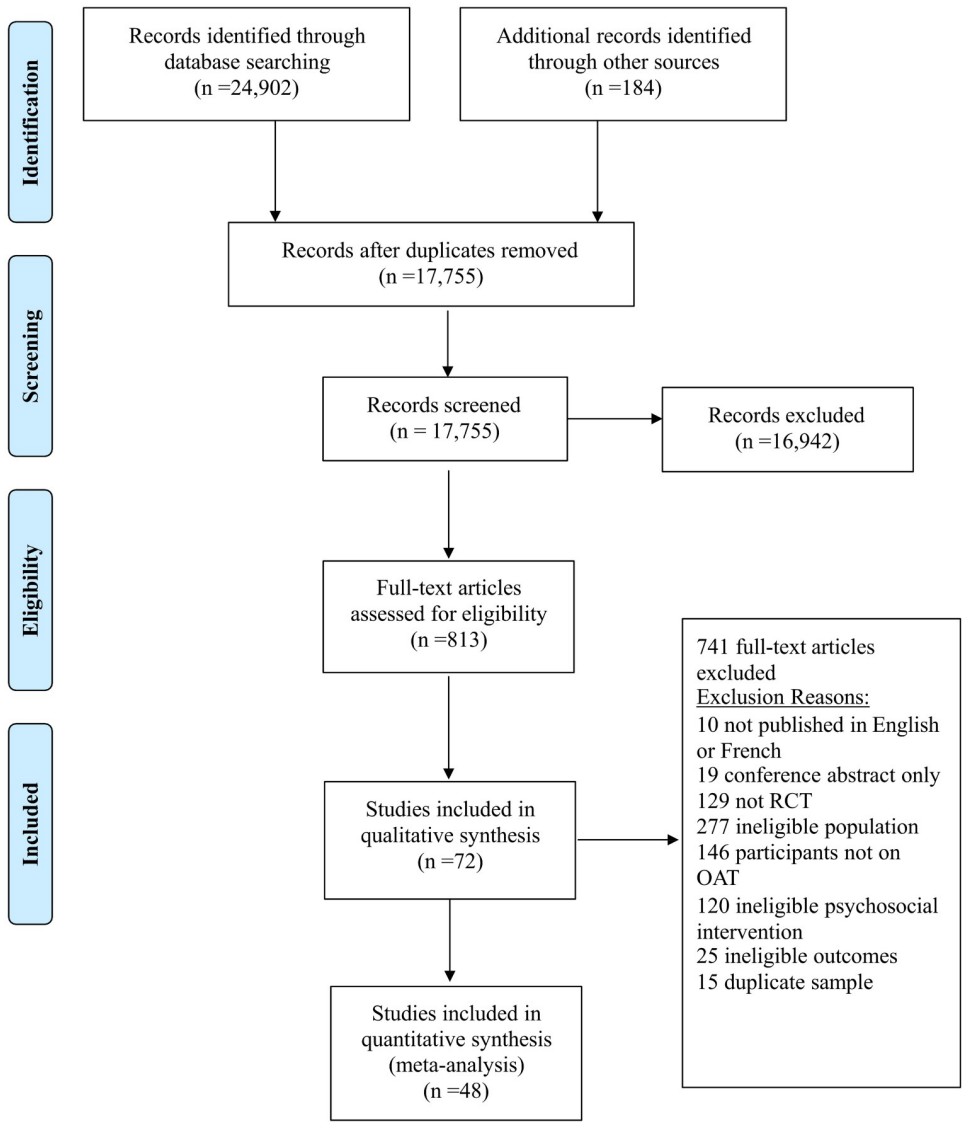

**Fig 1. Process of study selection.** A flow diagram is shown which depicts the process of study selection.

**Table 1. Overview of included studies.**

| First Author, Year | Journal | Country | No. of Participants (Randomized) | Age of all Participants (years), (mean (SD) and/or range) | Sex/Gender (men/male ((%)) | Majority Ethnicity/Race (%) | Type of Opioid Use | Comorbidities or Unique Patient Characteristics | DSM or ICD Diagnosis |
|---|---|---|---|---|---|---|---|---|---|
| **Abbott, 1998** | The American Journal of Drug and Alcohol Abuse | USA | 180 | 37 (9.1) | Men: 69% | Hispanic: 79% | NR | NA | Yes |
| **Abrahms, 1979** | The International Journal of the Addictions | USA | 15 | 28 (NR) / (23–37) | Male: 93% | African American: 50% | NR | NA | No |
| **Amini-Lari, 2017** | Iran Journal of Psychiatry and Behavioural Sciences | Iran | 118 | 38.6 (NR) | Men: 95% | Persian: 100% | NR | NA | Yes |
| **Avants, 1999** | American Journal of Psychiatry | USA | 291 | 36.8 (6.9) | Male: 70% | European Descent: 59% | Illicit | NA | Yes |
| **Ball, 2007** | Journal of Personality Disorders | USA | 30 | 37 (6.1) | Men: 50% | European Descent: 80% | Illicit | Participants with personality disorder | No |
| **Barry, 2019** | Drug and Alcohol Dependence | USA | 40 | 38.1 (11.3) | Male: 63% | European Descent: 85% | Prescription | Participants with back pain | Yes |
| **Bickel, 2008** | Experimental and Clinical Psychopharmacology | USA | 135 | 28.6 (NR) | Male: 56% | European Descent: 96% | NR | NA | Yes |
| **Brooner, 2007** | Drug and Alcohol Dependence | USA | 236 | 38.5 (7.9) | Male: 54% | European Descent: 35% | NR | NA | Yes |
| **Carroll, 1995** | The American Journal on Addictions | USA | 20 | 27.6 (NR) | Men: 0% | Nonminority: 79% | Illicit | Participants were pregnant | No |
| **Catalano, 1999** | Addiction | USA | 144 | 35.3 (5.8) | Male: 25% | European Descent: 77% | NR | NA | No |
| **Chawarski, 2011** | Drug and Alcohol Dependence | China | 37 | NR | NR | NR | NR | NA | No |
| **Chawarski, 2008** | Drug and Alcohol Dependence | Malaysia | 24 | 36.7 | Male: 81% | NR | Illicit | NA | Yes |
| **Chen, 2013** | Drug and Alcohol Dependence | China | 255 | 38.1 (5.7) | Male: 92% | NR | Illicit | NA | Yes |
| **Chopra, 2009** | Experimental and Clinical Psychopharmacology | USA | 127 | 31.8 (NR) | Male: 58% | European Descent: 98% | NR | NA | Yes |
| **Christensen, 2014** | Journal of Consulting and Clinical Psychology | USA | 170 | 34.3 (NR) / (20–63) | Male: 54% | European Descent: 95% | NR | NA | Yes |
| **Chutuape, 1999** | Drug and Alcohol Dependence | USA | 14 | 41 (NR) | Male: 79% | European Descent: 86% | Illicit | Participants with sedative/hypnotic dependence disorder | Yes |
| **Czuchry, 2009** | Journal of Psychoactive Drugs | USA | 82 | 40.4 (8.9) | Male: 70% | Hispanic: 63% | Illicit | NA | No |
| **Day, 2018** | BMC Psychiatry | UK | 83 | 37 (NR) / (25–61) | Male: 87% | European Descent 83% | Prescription and illicit | NA | No |
| **Downey, 2000** | Experimental and Clinical Psychopharmacology | USA | 41 | 40.0 (NR) | Male: 61% | Non-European Descent: 66% | Illicit | NA | Yes |
| **Epstein, 2009** | Drug and Alcohol Dependence | USA | 252 | 37.8 (7.6) / (19–57) | Men: 48% | African American: 66% | Illicit | NA | No |
| **Fals-Stewart, 2001** | Behavior Therapy | USA | 43 Couples | Male partner: 38.1 (7.5) / Female partner: 36.0 (7.3) | Male: 50% | Male: European Descent: 50% / Female: European Descent: 56% | Illicit | Heterosexual couples only | No |

*(Continued)*

**Table 1.** (Continued)

| First Author, Year | Journal | Country | No. of Participants (Randomized) | Age of all Participants (years), (mean (SD) and/or range) | Sex/ Gender (men/ male ((%)) | Majority Ethnicity/Race (%) | Type of Opioid Use | Comorbidities or Unique Patient Characteristics | DSM or ICD Diagnosis |
|---|---|---|---|---|---|---|---|---|---|
| **Fiellin, 2013** | American Journal of Medicine | USA | 141 | 33.7 (NR) | Male: 74% | European Descent: 90% | Prescription and illicit | NA | No |
| **Fiellin, 2006** | The New England Journal of Medicine | USA | 166 | 36.0 (NR) | Male: 78% | European Descent: 77% | Prescription and illicit | NA | Yes |
| **Ghitza, 2008** | Addictive Behaviors | USA | 116 | 37.0 (8.4) | Male: 56% | African American: 47% | Illicit | Participants were cocaine users | Yes |
| **Groß (Gross), 2006** | Experimental and Clinical Psychopharmacology | Germany | 60 | 32.5 (9.8) | Male: 55% | European Descent: 92% | Illicit | NA | Yes |
| **Gu, 2013** | AIDS Behavior | China | 288 | NR | Male: 92% | Han Chinese: 99% | NR | NA | No |
| **Hosseinzadeh, 2014** | Archives of Psychiatric Nursing | Iran | 35 | 29.5 (NR) / (17–43) | Male: 100% | NR | NR | Participants with depression | No |
| **Hser, 2011** | Addiction | China | 320 | 38 | Male: 77% | NR | Illicit | NA | NR |
| **Iguchi, 1997** | Journal of Consulting and Clinical Psychology | USA | 103 | 36.3 (6.9) | Male: 63% | European Descent: 85% | Illicit | NA | No |
| **Jaffray, 2014** | International Journal of Pharmacy Practice | Scotland | 542 | 32 (NR) | Male: 64% | NR | NR | NA | No |
| **Jiang, 2012** | Shanghai Archives of Psychiatry | China | 160 | 38.9 (8.9) | Male: 78% | Han Chinese: 98% | Illicit | NA | Yes |
| **Joe, 1997** | The Journal of Nervous and Mental Disease | USA | 454 | 36 (NR) | Men: 68% | Mexican American: 44% | NR | NA | No |
| **Karow, 2010** | Drug and Alcohol Dependence | Germany | 1015 | 36.4 (6.7) | Male: 80% | NR | Illicit | Participants were in poor physical and/or mental health* | Yes |
| **Kelly, 2012** | Journal of Addiction Medicine | USA | 244 | 43.2 (8.0) | Male: 70% | African American: 77% | Illicit | NA | No |
| **Kidorf, 2018** | Drug and Alcohol Dependence | USA | 212 | 39.8 (NR) | Male: 55% | European Descent: 37% | NR | NA | No |
| **Kosten, 2003** | Drug and Alcohol Dependence | USA | 160 | 36.9 (NR) | Male: 66% | European Descent: 56% | Illicit | Participants were cocaine users | Yes |
| **Linehan, 2002** | Drug and Alcohol Dependence | USA | 24 | 36.1 (7.3) | Men: 0% | European Descent: 66% | Illicit | Participants with borderline personality disorder | Yes |
| **Ling, 2013** | Addiction | USA | 202 | 37.0 (NR) | Male: 69% | European Descent: 52% | NR | NA | Yes |
| **Liu, 2018** | Frontiers in Psychiatry | China | 125 | 43.9 (6.6) | Male: 74% | NR | NR | NA | No |
| **Marsch, 2014** | Journal of Substance Abuse Treatment | USA | 160 | 40.7 (9.8) | Male: 75% | European Descent: 44% | NR | NA | Yes |
| **McLellan, 1993** | The Journal of the American Medical Association | USA | 102 | 41 (NR) | Male: 100% | African American: 74% | Illicit | Participants were all male veterans | Yes |
| **Milby, 1978** | Addictive Behaviors | USA | 75 | NR (NR)/(21–54) | Male: 83% | African American: 52% | NR | NA | No |
| **Miotto, 2012** | Journal of Addiction Medicine | USA | 104 | 35.4 (NR) | Male: 58% | European Descent: 58% | Prescription and illicit | NA | Yes |

(*Continued*)

**Table 1.** (Continued)

| First Author, Year | Journal | Country | No. of Participants (Randomized) | Age of all Participants (years), (mean (SD) and/or range) | Sex/ Gender (men/ male ((%)) | Majority Ethnicity/Race (%) | Type of Opioid Use | Comorbidities or Unique Patient Characteristics | DSM or ICD Diagnosis |
|---|---|---|---|---|---|---|---|---|---|
| **Moore, 2013** | Journal of Substance Abuse Treatment | USA | 36 | 41.3 (NR) | Male: 42% | European Descent: 59% | Prescription and illicit | NA | No |
| **Moore, 2019** | Journal of Substance Abuse Treatment | USA | 82 | 42.4 (NR) | Male: 60% | European Descent: 67% | Prescription and illicit | NA | No |
| **Nyamathi, 2011** | Journal of Addiction Diseases | USA | 256 | 51.2 (8.4) | Male: 59% | African American: 45% | NR | Participants with moderate-to-heavy alcohol use | No |
| **O'Connor, 1998** | The American Journal of Medicine | USA | 46 | 33.5 (NR) | Male: 69% | European Descent: 71% | Illicit | NA | Yes |
| **Oliveto, 2005** | Addiction | USA | 140 | 36.5 (NR) | Males: 68% | European Descent: 65% | Illicit | NA | Yes |
| **O'Neill, 1996** | Drug and Alcohol Dependence | Australia | 92 | 26.2 (4.5) | Men: 0% | NR | NR | All participants were pregnant | No |
| **Otto, 2014** | Journal of Psychoactive Drugs | USA | 78 | 42.3 (9.9) | Men: 55% | European Descent: 68% | Illicit | NA | Yes |
| **Pan, 2015** | PLoS ONE | China | 240 | 40.9 (8.5) | Male: 78% | Chinese: 100% | NR | NA | Yes |
| **Pashaei, 2013** | Iranian Journal of Public Health | Iran | 92 | 37.7 (10.9) | Male: 100% | NR | NR | NA | No |
| **Petry, 2002** | Journal of Consulting and Clinical Psychology | USA | 42 | 38.6 (NR) | Male: 29% | Hispanic: 52% | NR | Participants were cocaine users | Yes |
| **Poling, 2006** | Archives of General Psychiatry | USA | 106 | 34.6 (9.0) | Male: 70% | European Descent: 76% | NR | Participants were cocaine users | Yes |
| **Pollack, 2002** | Journal of Substance Abuse Treatment | USA | 23 | 41.0 (NR) | Men: 44% | European Descent: 78% | Illicit | NA | Yes |
| **Preston, 2002** | Drug and Alcohol Dependence | USA | 110 | 37.6 (NR) | Male: 69% | African-American: 39% | NR | NA | No |
| **Preston, 2000** | Archives of General Psychiatry | USA | 120 | 37.7 (NR) | Male: 68% | European Descent: 58% | Illicit | NA | No |
| **Rounsaville, 1983** | Archives of General Psychiatry | USA | 72 | NR | Male: 61% | European Descent: 58% | NR | Participants with a psychiatric disorder or personality disorder | No |
| **Rowan-Szal, 1997** | Journal of Maintenance in the Addictions | USA | 46 | 38 / (25–60) | Male: 91% | Mexican American: 58%, European Descent: 37% | Illicit | NA | No |
| **Salehi, 2018** | Shiraz E-Medical Journal | Iran | 50 | 36.0 (NR) | Male: 92% | NR | Illicit | NA | Yes |
| **Scherbaum, 2005** | European Addiction Research | Germany | 73 | 30 (6) / (19–41) | Male: 73% | German: 96% | NR | NA | Yes |
| **Schottenfeld, 2005** | American Journal of Psychiatry | USA | 163 | 36.2 (6.3) | Male: 66% | European Descent: 52% | Illicit | NA | Yes |
| **Schwartz, 2012** | Addiction | USA | 244 | 43.2 (8.0) | Male: 70% | African American: 77% | NR | NA | No |
| **Shi, 2020** | Substance Abuse | USA | 20 | 40.5 (12.2) | Male: 60% | European Descent: 100% | NR | NA | Yes |
| **Silverman, 2004** | Journal of Consulting and Clinical Psychology | USA | 78 | 39.1 (NR) | Male: 55% | African American: 69% | NR | NA | Yes |

(*Continued*)

**Table 1.** (Continued)

| First Author, Year | Journal | Country | No. of Participants (Randomized) | Age of all Participants (years), (mean (SD) and/or range) | Sex/ Gender (men/ male ((%)) | Majority Ethnicity/Race (%) | Type of Opioid Use | Comorbidities or Unique Patient Characteristics | DSM or ICD Diagnosis |
|---|---|---|---|---|---|---|---|---|---|
| **Stein, 2015** | Drug and Alcohol Dependence | USA | 49 | 41.1 (11.3) | Male: 65% | European Descent: 86% | NR | NA | No |
| **Sullivan, 2006** | Clinical Infectious Diseases | USA | 16 | 47.2 (8.5) | Male: 94% | African American: 44% | Illicit | Participants were HIV-positive | Yes |
| **Tetrault, 2012** | Journal of Substance Abuse Treatment | USA | 47 | 46.9 (8.0) | Male: 83% | European Descent: 57% | Illicit | Participants were HIV-positive | Yes |
| **Tuten, 2012** | The American Journal of Drug and Alcohol Abuse | USA | 143 | 30.0 (5.2) | Men: 0% | African American: 71% | Illicit | Participants were pregnant | Yes |
| **Woody, 1995** | American Journal of Psychiatry | USA | 123 | 41 (7.0) | Men: 60% | African American: 57% | NR | Participants with mid to high levels of psychiatric symptoms. | No |
| **Woody, 1987** | American Journal of Psychiatry | USA | 120 | NR | Male: 100% | NR | NR | Participants were male veterans | No |
| **Yaghubi, 2017** | Addiction and Health | Iran | 60 | 30.2 (NR) | Men: 100% | NR | NR | NA | Yes |

*Poor physical and/or mental health as defined by at least 13 symptoms on the Opiate Treatment Index Health Symptoms Scale or at least 60 points (standardized T-score) on the Global Severity Index of the Symptom Check-List (SCL-90-R).

NA = Not applicable; NR = Not reported.

9.7%) [42, 43, 48, 50, 59, 61, 105], Iran (n = 5, 6.9%) [41, 45, 96, 106, 107], Germany (n = 3, 4.2%) [62, 72, 81], the United Kingdom (n = 1, 1.4%) [40], Australia (n = 1, 1.4%) [91], Scotland (n = 1, 1.4%) [109], and Malaysia (n = 1, 1.4%) [60]. Most trials were funded by government or not-for-profit organization grants (n = 64, 88.9%) [39–44, 47–53, 55–59, 61, 62, 64–90, 92–99, 101–104, 106–110], and 11.1% (n = 8) of studies did not report a funding source [45, 46, 54, 60, 63, 91, 100, 107]. Most studies delivered interventions in the community (n = 60, 83.3%) [40–43, 45–50, 52, 54–65, 68–71, 73–80, 82, 83, 85, 86, 88–93, 95, 96, 98–111], with few studies providing interventions at a hospital (n = 4, 5.6%) [53, 72, 87, 97], a research clinic (n = 3, 4.2%) [66, 67, 81], an individual's own home (n = 2, 2.8%) [51, 84], or a Veterans' Affairs clinic (n = 1, 1.4%) [94]. The remaining studies did not report the environment in which the intervention was delivered (n = 2, 2.8%) [39, 44]. Four studies (5.6%) [41, 96, 106, 107] were published in journals that were open access but were not listed in the directory of open access journals, and were associated with characteristics of predatory journals (e.g., low fees for open access publications, spelling errors on journal websites).

## Interventions compared

Fig 2 presents the extent to which different psychosocial interventions were assessed across the set of included studies; a total of 35 different treatments among 150 study arms were eligible among the 72 included studies. The majority of studies included two eligible arms (n = 67, 93.1%) [39, 41–48, 50–89, 91, 93, 95–110], while a few studies included three (n = 4, 5.6%) [40, 90, 92, 94] or four (n = 1, 1.4%) [49] eligible arms. In line with our inclusion criteria, all eligible study arms included OAT. The most commonly reported psychosocial interventions among study arms used in addition to OAT were counselling plus CM (n = 17/150 arms, 11.3%) [49,

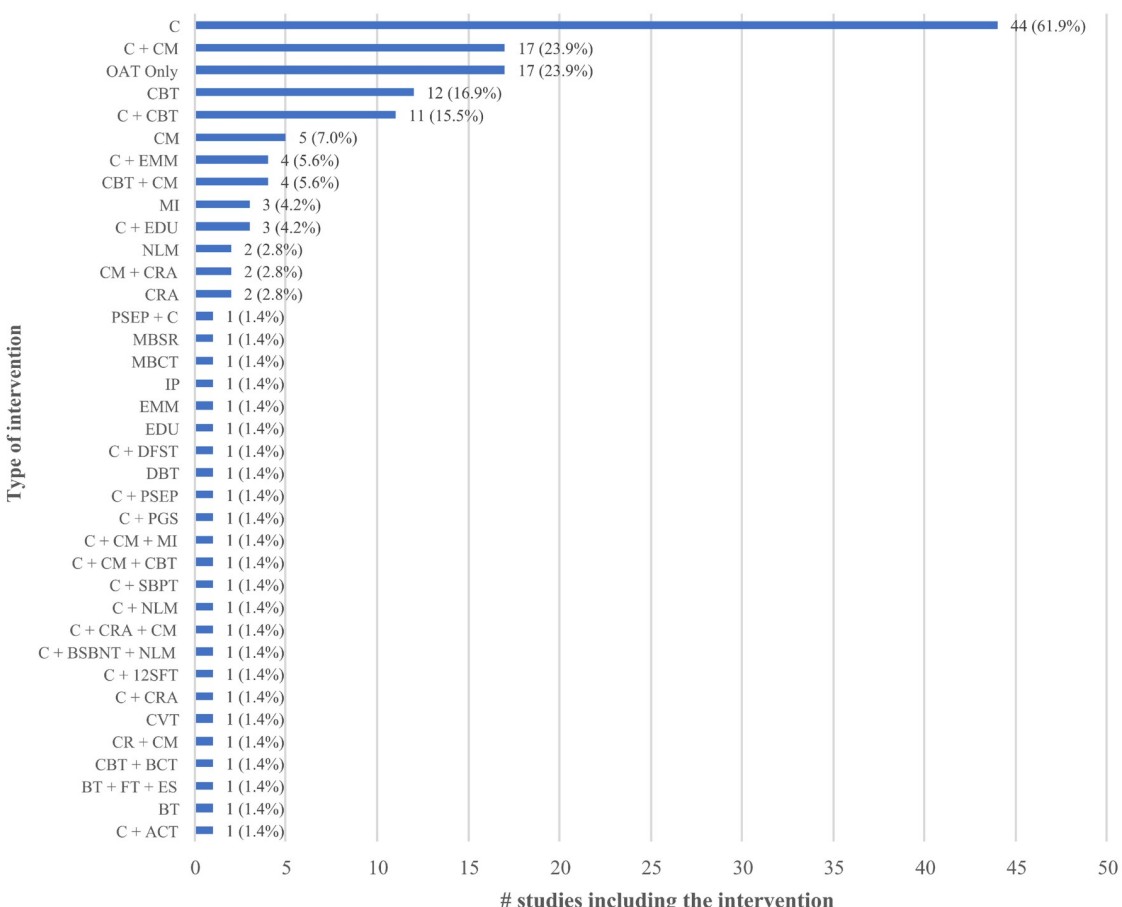

**Fig 2. Type of psychosocial interventions.** The number and type of different psychosocial interventions that were assessed across the set of included studies is presented.

57, 58, 65, 67, 73, 75, 77, 78, 80, 81, 83, 85, 98, 101, 104, 108] and counselling plus CBT (n = 11/ 150 arms, 7.3%) [43, 47, 49, 51, 52, 76, 86, 91, 92, 102, 107], while the most common control groups were counselling in addition to OAT (n = 44/72 control groups, 62.5%) [40, 42, 46, 47, 49, 51–54, 56, 60, 63, 65, 67, 70, 72, 73, 75, 78, 80, 81, 83–93, 95, 97, 98, 101–104, 107], OAT-only (n = 17/74 control groups, 23.0%) [41, 43, 45, 48, 50, 55, 57–59, 61, 94, 96, 100, 105, 106, 109]. Less common intervention characteristics were found in two studies (2.8%) that delivered a telephone-based intervention [51, 102], one study that recruited couples (1.4%) [99], and one study that randomized community pharmacies and had pharmacists deliver motivational interviewing (1.4%) [109]. Two studies (2.8%) delivered interventions through a stepped care model (see S1 Data) [69, 104].

## Study populations

The majority of studies were comprised of more men than women (median = 69% male) [39–50, 52–54, 56–60, 62–64, 66, 67, 69–75, 78, 81–83, 85–90, 92–97, 100–110, 112], with 24 studies (32.3%) having a sample comprising more than 75% males [40–43, 45, 48, 50, 53, 54, 59, 60, 62, 70, 85, 92, 94–97, 100–102, 106, 107]. The median of mean participant ages was 37.3 years (range of mean age from 26.2 to 51.2 years). Approximately half of all studies (51.4%) reported

a focus on individuals using drugs intravenously [39, 42, 43, 48, 51–57, 59, 61–63, 65–68, 70, 73–75, 78, 80–89, 91, 94, 99, 101, 104, 108]. The duration of opioid use was inconsistently reported in studies, with reporting formats including age first using opioids, mean use of opioids in the past year, use of opioids in the past week, self-reported use in the past year, among other formats. Overall, 53 of 72 studies (73.6%) included individuals who were receiving methadone [39–43, 45–48, 50, 51, 55, 57–59, 61–63, 65, 67, 69, 71, 72, 75, 77, 78, 80, 83–86, 88–107, 109], while the remainder included individuals who were receiving buprenorphine/naloxone (n = 13, 18.1%) [44, 49, 52–54, 56, 60, 64, 66, 68, 70, 74, 82, 108], buprenorphine (n = 4, 5.6%) [76, 81, 87, 110], levacetylmethadol (LAAM; n = 2, 2.7%) [73, 79].

A subset of studies focused on sub-populations of individuals receiving OAT that (a) had an additional substance use condition [n = 6, 8.3%; these included cocaine use (n = 4, 5.6%) [67, 71, 76, 80], moderate to heavy alcohol use (n = 1, 1.4%) [39], or sedative/hypnotic dependence (n = 1, 1.4%) [85], (b) had a psychiatric disorder or prominent symptoms associated with a psychiatric disorder [n = 5, 6.9%, including a personality disorder (n = 2, 2.8%) [68, 79], personality or psychiatric disorder (n = 1, 1.4%) [95], depression (n = 1, 1.4%) [45], or mid- to high level psychiatric symptoms (n = 1, 1.4%) [93], (c) were pregnant women (n = 3, 4.2%) [55, 91, 98], (d) were veterans (n = 2, 2.8%) [92, 94], (e) were positive for HIV (n = 2, 2.8%) [54, 70], (f) had chronic low back pain (n = 1, 1.4%) [103] or (g) were described as being in poor mental or physical health based on a predetermined cutoff on the Opiate Treatment Index Health Symptoms Scale of the Global Severity Index of the Symptom Check-list (n = 1, 1.4%) [62]. One study only recruited individuals that had "failed to respond to the standard course of treatment" [46].

## Outcomes reported

Fig 3 presents the extent to which the outcomes of a priori interest were reported for each eligible treatment arm across the set of included studies. The most commonly reported outcome measures were treatment retention (81.9% of studies) [40, 42–44, 46–60, 64–66, 68–83, 86, 87, 89–91, 93, 95–99, 101–105, 107–110], abstinence from opioids (48.6%) [40, 42–44, 47–49, 51–56, 59, 60, 64, 65, 69, 71, 73–76, 78–80, 82, 83, 86–88, 94, 103, 105, 110], drug use (47.2%) [39,

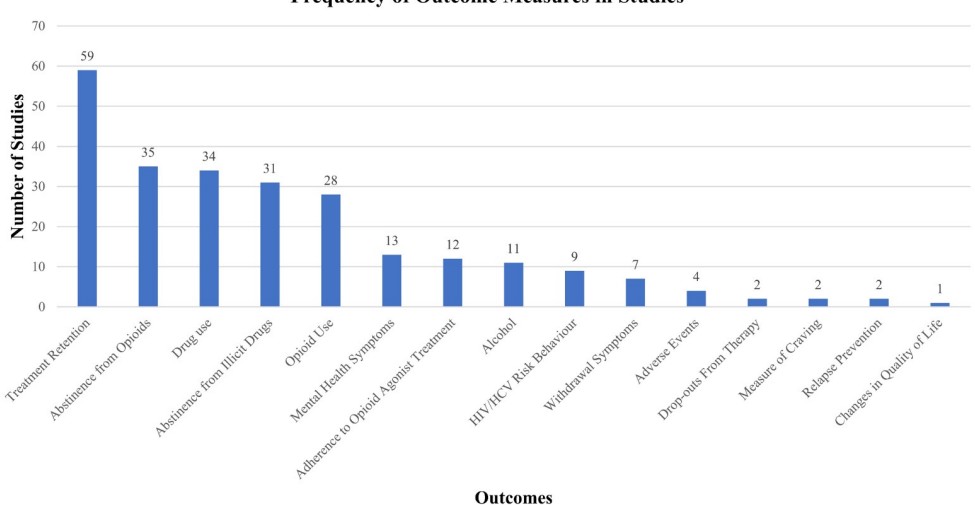

**Fig 3. Outcomes reported in included studies.** The extent to which the outcomes of a priori interest were reported for each eligible treatment arm across the set of included studies are presented.

42, 46, 49, 53, 58, 63, 64, 68, 71, 72, 75, 77, 79–82, 84, 85, 87–89, 91–95, 97–99, 101, 104, 108, 109], abstinence from illicit drugs (43.1%) [51–55, 64–67, 69, 71, 73–83, 85, 86, 90, 94, 100, 102, 107, 108], and opioid use [38.8%; including urinalysis (23.6%) [44, 49, 58, 61, 63, 65, 70–72, 79, 87, 89, 93, 94, 98, 99, 101, 104, 106] and self-reported use (15.3%)] [42, 51–53, 59, 78, 83, 84, 92, 96, 109], respectively. Aside from these five outcomes, all others, including mental health symptoms [40, 42, 43, 45, 68, 71, 73, 76, 77, 88, 92, 95, 97], alcohol use [42, 43, 58, 80–83, 88, 92, 95, 104], adherence to OAT [42, 43, 49, 53, 54, 61, 72, 86, 102, 110], HIV/HCV risk behavior [54, 57, 60, 61, 67, 88, 89, 91, 105], withdrawal symptoms [40, 49, 65, 73, 86, 87, 96], adverse events [49, 58, 75, 102], measure of craving [49, 83], relapse prevention [41, 106], drop-outs from therapy [46, 95], and quality of life [62] were reported in fewer than 15 studies (minimum = 1 study, maximum = 13 studies) (see S6 Text). In addition to sparse reporting of many of the outcomes of interest, considerable heterogeneity was identified in terms of how outcomes were defined and formatted for reporting (see S8 through S24 Texts). Studies also varied in follow-up timepoints with as few as four weeks [51, 76] and as many as 64 months of follow-up [79]. However, the majority of follow-up time points were 12 or 24 weeks (median = 24 weeks, interquartile range = 13).

## Risk of bias of the included evidence

Evaluations using the Cochrane RoB tool identified several limitations of the evidence base. A total of 58 studies (80.6%) [39, 41, 42, 44, 45, 47–49, 51, 53–56, 59, 60, 62–64, 66–74, 76–81, 84–102, 104–106, 109, 110] failed to provide details of the methods used for concealment of randomization allocation resulting in an unclear RoB for this item. When comparing published reports to registered protocols (where available) and methods sections, there was a high or unclear RoB for selective outcome reporting in 83.3% (n = 60) [40, 42–44, 46–51, 53–61, 63–84, 86–99, 101, 107–110] of trials for at least one critical outcome reviewed. Objective measures were used for all studies that measured treatment retention (n = 59/59, 100%) [40, 42–44, 46–60, 64–66, 68–83, 86, 87, 89–91, 93, 95–99, 101–105, 107–110], and most that measured opioid use (n = 46/51, 90.2%) [42–44, 47–49, 51–56, 58–61, 63–65, 69–76, 78–80, 82, 83, 86–89, 93, 94, 98, 99, 101, 103–105, 107, 110] and adherence to OAT (n = 10/12, 83.3%) [42, 43, 48–50, 53, 54, 61, 72, 102, 110], suggesting a low risk of detection bias for these outcomes. Of the four studies measuring adverse events, only one had a low risk of detection bias (n = 1/4, 25.0%) [58]. Given the nature of psychological interventions, however, all included studies (100.0%) [39–110] had a high risk of bias due to the inability to blind participants and clinicians to the delivery of psychological interventions. See S7 Text for a study-by-study account of the evaluations of RoB.

## Syntheses for primary outcomes

Following inspection of the availability of outcomes across studies, including both outcome type (e.g., 'opioid use', 'opioid abstinence', 'retention') and approach to measurement (e.g., numbers of days of abstinence versus abstinence beyond 90 days; number of therapy sessions attended versus number of individuals attending 80% or more of sessions); participant population characteristics and study methods, NMA was unlikely to produce reliable findings for one co-primary outcome (opioid use, including opioid abstinence) and all secondary outcomes. Only treatment retention, measured as a dichotomous endpoint, was analyzed using NMA.

**Findings, treatment retention.** Fifty-nine studies reported treatment retention data [40, 42–44, 46–60, 64–66, 68–83, 86, 87, 89–91, 93, 95–99, 101–105, 107–110]. Forty-eight of these studies, representing data from 5,404 participants and 20 unique interventions, were included in an NMA using the most universally reported format of retention data, which was the

number of individuals retained in treatment at the study's latest follow-up time [40, 42–44, 46–60, 64–66, 69–75, 79–83, 86, 87, 90, 91, 95, 96, 99, 102–105, 107–109]. In addition to the 48 included studies, one additional study was eligible for inclusion in the NMA but was disconnected from the evidence network due to the interventions not being tested in any other studies (i.e., counselling plus education versus counselling plus ACT); its findings are reported descriptively in S8 Text [44]. Fig 4 presents a network diagram summarizing the available evidence used for the NMA. The primary analysis was based upon an unadjusted random effects NMA model, which fit the data well based upon assessment of model fit statistics (see S27 Text for numeric details, including evaluation of the consistency assumption and a comparison adjusted funnel plot).

CBT plus behavioural couples therapy (one study, 21 participants) was the highest ranked treatment based on a SUCRA value of 0.85. The next highest ranked interventions were counselling plus CM plus community reinforcement approach (0.82; one study, 92 participants), counselling plus personal goal setting (0.80; one study, 83 participants), CM (0.88; five studies, 414 participants), and CBT plus CM (0.63; one study, 94 participants). While SUCRA values provide insight as to differential rates of retention between psychosocial interventions, pairwise comparisons from NMA (Fig 5) suggest that counselling plus CM plus community reinforcement approach (OR 2.79, 95% CrI 1.09–7.23) and CM (OR 2.01, 95% CrI 1.28–3.01) each resulted in significantly greater treatment retention as compared to OAT-only. Statistically

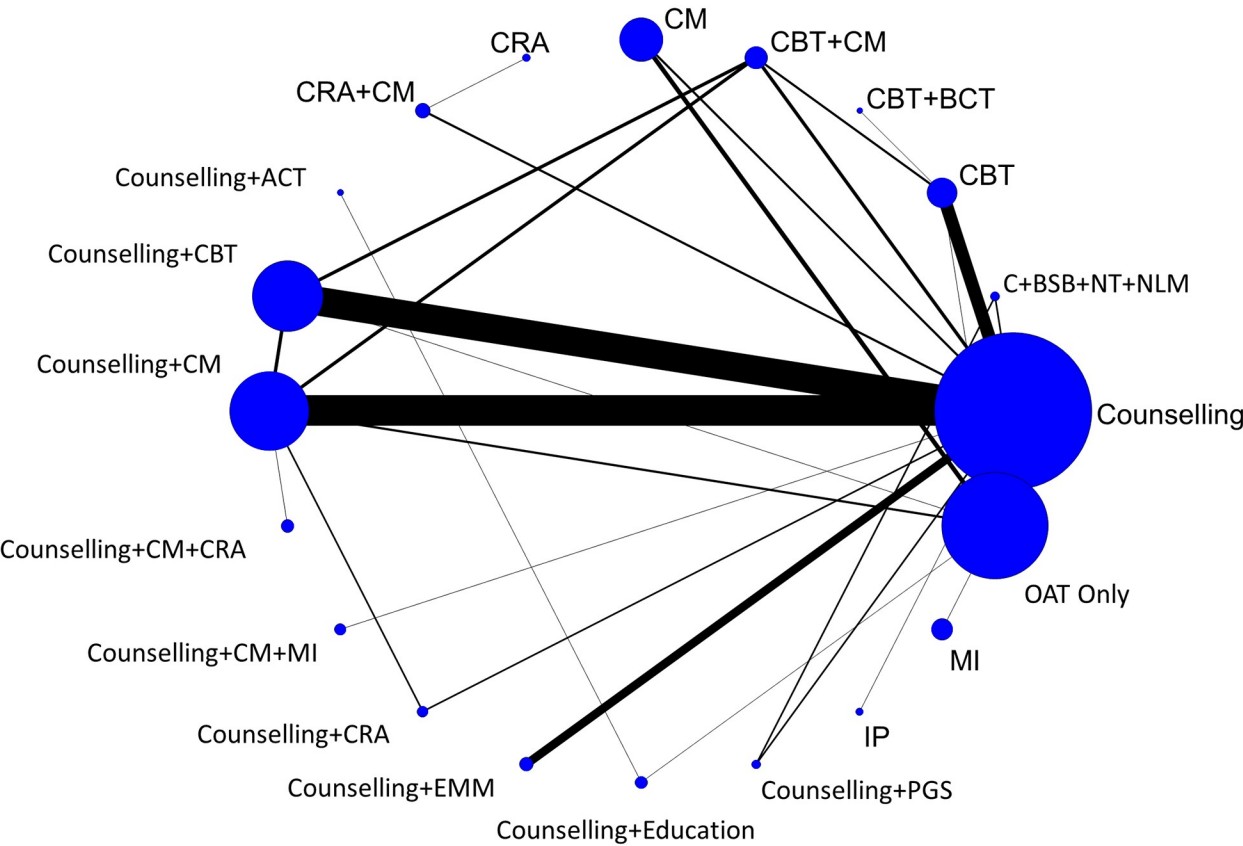

**Fig 4. Network diagram of treatment retention.** The evidence network of the available studies and interventions for treatment retention as a binary outcome is shown. Joining lines denote treatment comparisons where one or more trials were available. Nodes are proportionally sized to reflect the numbers of patients studied with each intervention. Edge width reflects the number of RCTs for each comparison.

**Comparison vs OAT Only**

**OR (95% CrI); SUCRA**

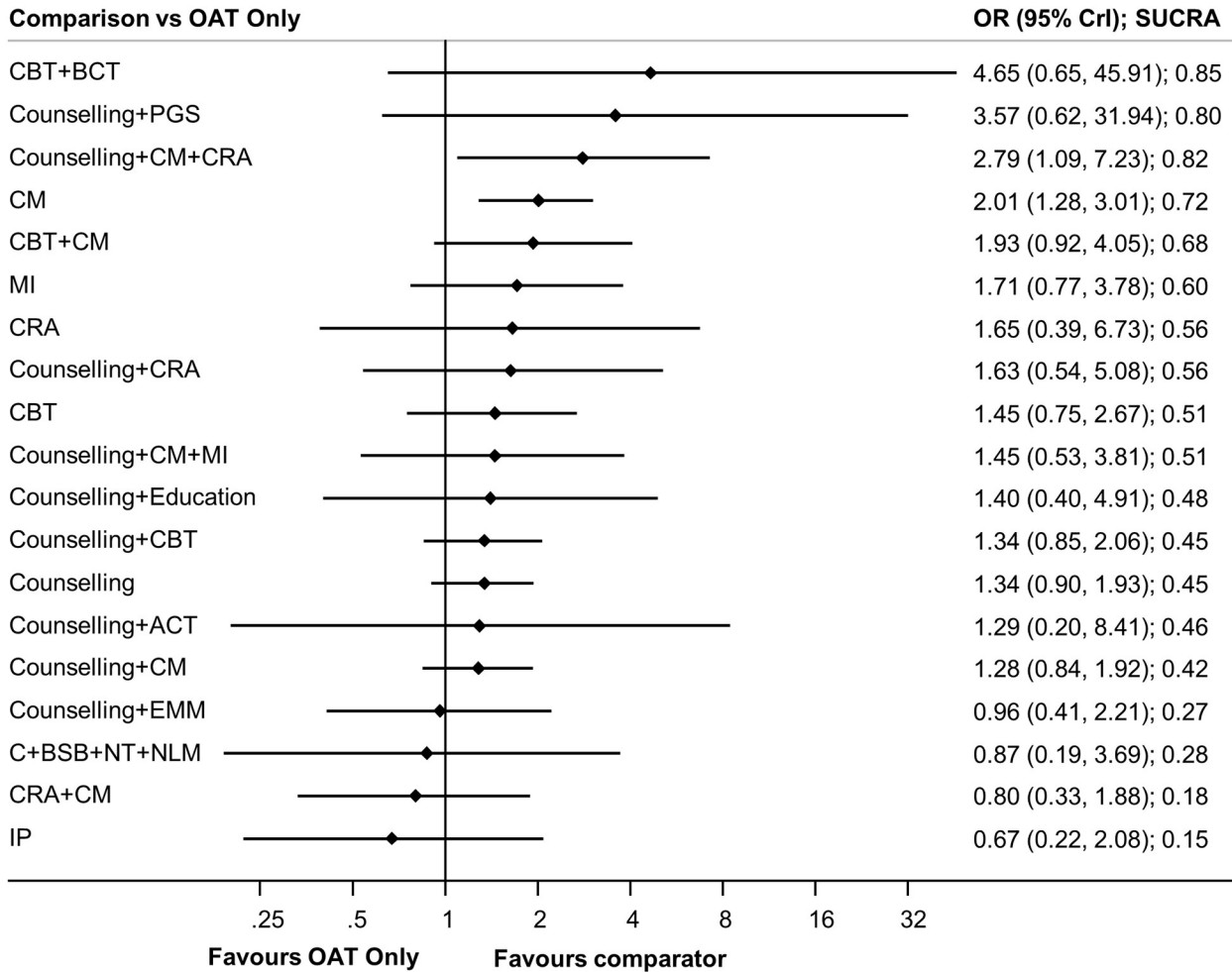

Fig 5. **Forest plots of treatment retention.** Forest plots of psychosocial treatments versus the reference group, OAT only, is presented.

significant differences were also found favouring counselling plus CM plus community reinforcement approach (OR 3.46, 95% CrI 1.05–11.23) and CM (OR 2.50, 95% CrI 1.00–6.30) as compared to CM plus community reinforcement approach, and when counselling plus CM plus community reinforcement approach (OR 4.19, 95% CrI 1.03–17.19) was compared to interpersonal psychotherapy. Amongst all other pairwise comparisons, no statistically significant differences were identified (see league table provided in S28 Text).

Secondary analyses involving NMA-based univariate meta-regression analyses that adjusted for cross-study variability in control group event rates, average age, sex (% males), and follow-up duration (number of weeks of follow-up per study) results remained similar to those observed in the unadjusted analysis (see S27 Text). Assessment of the comparison adjusted funnel plot identified no signs of publication bias (see S27 Text). The ten studies [76–78, 89, 93, 97, 98, 101, 110, 113] which reported a different format for treatment retention and the one study [44] that was disconnected from the NMA are described in S8 Text.

**Findings, opioid abstinence.** The co-primary outcome of opioid use based on urinalysis was reported as both (1) abstinence from opioids and (2) opioid use, based on the description that authors provided of the outcome being opioid abstinence or use. A meta-analysis was not pursued for this outcome due to extensive heterogeneity in reporting formats; data are detailed next.

Thirty-five studies (48.6%) assessed abstinence from opioids through urinalysis [40, 42–44, 47–49, 51–56, 59, 60, 64, 65, 69, 71, 73–76, 78–80, 82, 83, 86–88, 94, 103, 105, 110]. Most studies (29/35, 82.9%) [40, 42, 43, 47–49, 51–56, 59, 60, 64, 65, 69, 73, 75, 78, 80, 83, 86–88, 94, 103, 105] compared a counselling [51, 53, 60, 65, 69, 73, 75, 76, 80, 86] or OAT- only [43, 48, 55, 59, 94, 105, 110] control group to a psychosocial intervention. Reporting formats regarding this outcome were heterogeneous and included the following: mean percentage of negative urinalyses for opioids through the duration of the study (n = 11) [48, 51, 53, 64, 69, 73, 75, 76, 80, 86, 110], mean maximum number of consecutive urine samples abstinent from opioids (n = 3) [42, 71, 103], and consecutive urine samples abstinent from opioids (n = 2) [87, 94], amongst other formats (see S9 Text for details). Thirty-three studies [40, 42, 43, 47–49, 51–56, 59, 60, 64, 65, 69, 71, 73–76, 78, 80, 82, 83, 86–88, 94, 103, 105, 110] included statistical significance results (i.e., p-values). Among these 33 studies, the majority (72.7%) found no significant difference between the treatment and control groups (n = 24/33) [40, 42, 49, 51–56, 59, 65, 69, 73, 74, 76, 78, 80, 82, 83, 86, 88, 103, 105, 110]. No patterns for interventions that consistently resulted in significant or non-significant findings as compared to control groups were found. Findings did not substantively differ due to the control group (see S9 Text).

**Findings, opioid use.** Seventeen studies (23.9%) [44, 49, 58, 61, 63, 70–72, 79, 87, 89, 93, 94, 98, 99, 101, 104] assessed opioid use through urinalysis, while an additional 11 studies [42, 51–53, 59, 78, 83, 84, 92, 96, 109] assessed opioid use through self-report methods only (see "self-reported opioid use" under secondary outcomes). Formats of reporting the use of opioids through urinalysis included the mean proportion of urine samples positive for opioids (n = 11) [49, 58, 61, 63, 71, 72, 87, 89, 94, 98, 101], the percent and associated number of urinalyses positive for opioids (n = 3) [44, 70, 104], and the number of participants with a positive urinalysis at final timepoint (n = 2) [79, 93], among other formats (see S10 Text for details). Of the 17 studies reporting opioid use based on urinalysis, 11 (64.7%) [44, 49, 58, 63, 70–72, 93, 98, 101, 104] found no statistically significant difference between the treatment and control conditions for opioid use. The six remaining studies (35.3%) [61, 79, 87, 89, 94, 99] reported a significantly greater reduction in opioid use for the intervention group as compared to the control group and included significant improvements after receiving dialectical behaviour therapy [79], CBT [87], CBT plus behavioural couples therapy [99], node link mapping [89], behavioural therapy plus family therapy plus employment services [114], and enhanced medical management [115]. A non-significant difference was consistently found when intervention groups were compared to counselling control groups or active control groups; however, two [61, 94] of three studies [58, 61, 94] (66.7%) that compared a psychological intervention to OAT-only significantly favoured the intervention group (see S10 Text).

### Syntheses for secondary outcomes

Details of all secondary outcomes are presented next. Outcomes are ordered from those reported in most to the fewest studies.

**Findings, drug use.** Thirty-four studies (47.2%) [39, 42, 46, 49, 53, 58, 63, 64, 68, 71, 72, 75, 77, 79–82, 84, 85, 87–89, 91–95, 97–99, 101, 104, 108, 109] assessed the use of illicit drugs (as described by the study authors) either through urinalysis (n = 18/34, 52.9%) [53, 58, 63, 71, 72, 79, 80, 85, 87, 89, 93–95, 97, 98, 101, 104, 109] or through self-report measures (n = 16/34, 47.0%) [39, 42, 46, 49, 64, 68, 75, 77, 81, 82, 84, 88, 91, 92, 99, 108]. Most studies (n = 21/34, 61.8%) [42, 46, 49, 53, 58, 63, 64, 72, 75, 80, 81, 84, 85, 87–89, 91–95, 97, 98, 101, 104, 109] were designed with a counselling [53, 63, 72, 80, 85, 87, 89, 93–95, 97, 98, 101, 104] or OAT-only [58, 94, 109] control group. Reporting formats were heterogeneous and included addiction severity index (n = 14) [39, 42, 46, 49, 64, 75, 77, 81, 82, 88, 92, 99, 108, 113], the mean percent of

urinalyses positive for cocaine (n = 5) [58, 63, 72, 89, 98], and self-reported use of cocaine in the past month (n = 2) [84, 91], among other formats (see S11 and S12 Texts for details of the specific drugs measured and detailed findings). Thirty-one studies reported only p-values associated with statistical tests to inform treatment comparisons [39, 42, 46, 49, 53, 58, 63, 64, 68, 71, 72, 75, 77, 79, 81, 82, 84, 88, 89, 91–95, 97–99, 101, 104, 108, 109] and the majority of these studies found no significant differences between treatment and control groups for drug use (n = 23/31, 74.2%) [39, 42, 46, 49, 53, 58, 63, 64, 68, 75, 77, 79, 81, 82, 89, 91, 92, 94, 95, 97, 98, 108, 109]. The remaining eight studies [71, 72, 84, 88, 93, 99, 101, 104] reported a statistically significant benefit of the intervention group as compared to the control condition (n = 8/31, 25.8%), with half (50.0%) [71, 88, 101, 104] of the significant intervention groups including a rewards-based component (i.e., CM or community reinforcement approach), while the remaining intervention groups reporting substantive improvements included counselling and psychoanalytic supportive-expressive psychotherapy, CBT, CBT plus behavioural couples therapy, and counselling plus skills-based parental training (see S11 and S12 Texts).

**Findings, abstinence from illicit drugs.** Thirty-one studies (43.1%) assessed abstinence from illicit drugs through urinalysis (n = 30/30, 100.0%) [51–55, 64–67, 69, 71, 73–83, 85, 86, 90, 94, 100, 102, 107, 108, 110]. Most studies (n = 24/31, 77.4%) [51–55, 64–66, 69, 73, 75, 77, 78, 80, 81, 83, 85, 86, 90, 94, 100, 102, 107] compared a counselling [51–54, 64–66, 69, 73, 75, 77, 78, 80, 81, 83, 85, 86, 90, 102, 107] or OAT-only control group [55, 94, 100, 110] to a psychosocial intervention, while the remaining studies compared to an active control group such as comprehensive validation therapy [79]. Reporting formats were highly heterogeneous and included the following: mean longest number of consecutive weeks abstinent from drugs (n = 8) [52, 64–66, 74, 75, 81, 85], mean weeks of continuous abstinence (n = 6) [53, 54, 76, 78, 79, 83], and mean percent of urinalyses negative from drugs (n = 6) [67, 69, 77, 80, 82, 110], among other formats (see S13 Text for details). Twenty-nine studies reported p-values associated with statistical tests to inform treatment comparisons [51–55, 64–66, 69, 71, 73–78, 80–83, 85, 86, 90, 94, 100, 102, 107, 108, 110], where findings were mixed, with 51.7% (n = 15/29) [51–55, 71, 74, 76, 78, 81, 82, 86, 90, 94, 102] of studies finding no statistically significant difference between groups for abstinence from illicit drugs and 50.0% (n = 14/28) [64–66, 69, 73, 75, 77, 80, 83, 85, 100, 107, 108, 110] finding a significantly greater benefit among the intervention group as compared to the control group. Twelve of 14 studies with statistically significant improvements as compared to the control group included CM as an adjunct to OAT (either alone, with counselling, or with community reinforcement approach), and 61.1% (n = 11/18) of studies that included a CM group reported significantly greater abstinence from illicit drugs as compared to an active or inactive control group. Findings were generally mixed irrespective of the type of control group. (see S13 Text).

## Additional secondary outcomes

The remaining secondary outcomes, including mental health symptoms [40, 42, 43, 45, 68, 71, 73, 76, 77, 88, 92, 95, 97], alcohol use [42, 43, 58, 80–83, 88, 92, 95, 104], adherence to OAT [42, 43, 48–50, 53, 54, 61, 72, 86, 102, 110], self-reported opioid use [42, 51–53, 59, 78, 83, 84, 92, 96, 109], HIV/HCV risk behavior [54, 57, 60, 61, 67, 88, 89, 91, 105], withdrawal symptoms [40, 49, 65, 73, 86, 87, 96], adverse events [49, 58, 75, 102], drop-outs from therapy [46, 95], measures of craving [49, 83], relapse [41, 106], and quality of life [62] were reported by fewer than 15 studies. Meta-analyses were not possible due to inconsistent reporting formats and the low frequency of studies reporting these outcomes. Table 2 provides an abbreviated summary of findings for each outcome for readers; we have also developed more detailed narrative summaries for readers (S29 Text), and study-by-study findings including intervention and control

**Table 2. Overview of findings for secondary outcomes with 15 or fewer studies.**

| Outcome | # studies reporting | Treatments represented in 1 or more RCTs | Diversity of comparators? | Assessments differed between studies? | Studies with p-values only? | Synopsis of Clinical Findings |
|---|---|---|---|---|---|---|
| Mental health symptoms | 13 [40, 42, 43, 45, 68, 71, 73, 76, 77, 88, 92, 95, 97] (n = 985) | C; CBT; CM; C + PSEP; C + CBT; C + CM; CM + CBT; CRA; IPT; C + CBT + CM; OAT only; MBCT; C + 12SFT; C + DFST; C + CM + MI; PGS; C + BSBNT + NLM | X | X | X | Of 13 studies that measured mental health symptoms, 9 reported on depression [71, 73, 76, 77, 92, 95, 111], 1 reported on distress [113], and 3 reported on general psychological functioning and psychiatric symptoms [43, 116, 117]. In these studies, only 2, both of which measured symptoms of depression, found a significant improvement in symptoms for CBT(alone or with counselling) [92, 111] and C+PSEP [92] as compared to C. |
| Alcohol use | 11 [42, 43, 58, 80–83, 88, 92, 95, 104] (n = 1149) | C; CBT; CRA; IPT; C + CBT; C + CM; CBT + CM; C + PSEP; C + CM + MI; OAT only | X | X | X | Of 11 studies that measured alcohol use in terms of abstinence or alcohol-related addiction severity index, only 1 study reported a difference between groups. The study compared C + CBT and C + PSEP with C, finding reduced alcohol in both groups compared to C [92]. |
| Adherence to OAT | 12 [42, 43, 48–50, 53, 54, 61, 72, 86, 102, 110] (n = 1840) | C; CM; CBT; C + CBT; C + CM; CBT + CM; C + CM + MI; EMM; C + EMM; CM + EMM; OAT only | X | X | | Of 12 studies that measured adherence to OAT, 2 studies reported differences [50, 118]. C [50] and CM [118] resulted in significantly greater adherence to OAT than OAT only control groups. |
| Self-reported opioid use | 11 [42, 51–53, 59, 78, 83, 84, 92, 96, 109] (n = 1542) | C; CM; C + PSEP; C + CBT; MI; C + SBPT; C + CM + MI; C + EMM; OAT only | X | X | X | Of 11 studies that measured self-reported opioid use, 3 studies reported differences [78, 92, 96]. CM (with counselling) [78], CBT (alone or with counselling) [92, 96], and skills-based parental training [92] resulted in significantly less self-reported opioid use as compared to the control group. |
| HIV/HCV risk behavior | 9 [54, 57, 60, 61, 67, 88, 89, 91, 105] (n = 1085) | C; C + CBT; C + CM; NLM; OAT only; EMM; CRA; C + EMM; C + Ed | X | X | X | In 3 studies evaluating risk behavior related to drugs [67, 89, 119], all found no important differences between groups (C vs CBT, 1 study; C vs C+CM, 1 study; C vs NLM, 1 study). |
| | | | | | | In 2 studies evaluating risk behavior related to sex 1 found increased reduction in risky behavior with C +CM compared to C. |
| | | | | | | In 6 studies evaluating risk behavior related to drugs and sex, reductions in risky behavior were noted by one study of EMM vs C and one study of C + EMM vs C. |

(*Continued*)

**Table 2.** (*Continued*)

| Outcome | # studies reporting | Treatments represented in 1 or more RCTs | Diversity of comparators? | Assessments differed between studies? | Studies with p-values only? | Synopsis of Clinical Findings |
|---|---|---|---|---|---|---|
| Withdrawal symptoms | 7 [40, 49, 65, 73, 86, 87, 96] (n = 677) | C; CBT; C + CM; C + CBT; CBT + CM; PGS; C + BSBNT + NLM; OAT only | X | X | X | No study reported the presence of important differences between comparators. Study comparisons involved C vs C+CM (2 studies) [65, 73]; OAT only vs CBT (1 study) [96]; C + CBT vs C + CBT + CM (1 study) [76]; C vs C + CBT vs C+CM vs CBT +CM (1 study) [49]; C vs CBT (1 study) [87]; C vs PGS vs C+BSBNT+NLM (1 study). [40] |
| Adverse events | 4 [49, 58, 75, 102] (n = 539) | C; OAT; C+CBT; C+CM; CBT+CM | X | X | | Formal comparisons were not reported. Authors concluded no substantive differences between intervention strategies in all studies [49, 58, 75, 102]. |
| Dropout from psychotherapy | 2 [46, 95] (n = 128) | C; CBT; IP | X | | | IPT was associated with fewer dropouts than C (1 study) [95]; no differences between C and CBT were observed (1 study). [46] |
| Cravings | 2 [49, 83] (n = 259) | C; C+CBT; C+CM; CM+CBT | X | X | | There was insufficient evidence to identify important differences in cravings between C, C+CBT, C+CM and CBT+CM (1 study) [49]; or between C+CM (1 study) [83] |
| Relapse | 2 [41, 106] (n = 152) | CBT; MBSR; OAT only | X | X | | Individuals receiving CBT (1 study) [41] and MBSR (1 study) relapsed less frequently than those receiving OAT only [106] |
| Quality of life | 1 [62] (n = 455) | MI; C + Psych-ed | X | | | No difference between MI and C + Psych-Ed was found [62]. |

Brief summaries of findings for outcomes with information from fewer than fifteen studies are provided. Detailed synopses for each outcome are provided in Appendix S29 Text, with study-level data provided within Appendices S14–S24 Texts. Challenges to the performance of meta-analyses are also indicated for each outcome measure. Diversity of comparators was considered a barrier when disconnected networks of evidence were present and/or treatment comparisons were largely informed by single studies. Differences between studies in assessment related to variations in endpoint definition, measurement scales used and/or timing of measurement. Studies reporting only p-values associated with findings from between-group comparisons were noted. Based upon these considerations as well as others related to clinical heterogeneity of patient populations and study methods, certain outcomes were not considered amenable to meta-analyses that would be meaningful for end users.

KEY: BSBNT = brief social behavior and network therapy; C = counselling; CBT = cognitive behavioral therapy; CM = contingency management; CRA = community reinforcement approach; EDU = education; EMM = enhanced medical management; IPT = interpersonal psychotherapy; MI = motivational interviewing; NLM = node link mapping; OAT = opioid agonist therapy; PGS = personal goal setting; 12SFT = 12-step facilitation therapy.

group details, follow-up time, and description of outcome measures are reported in S8 through S24 Texts.

## Specialty populations

While a priori sensitivity analyses were planned for specific populations (e.g., pregnant women, youths, incarcerated individuals) and treatment levels (e.g., individual, family, couples groups), none of these additional analyses could be performed as a consequence of sparse reporting of subgroup information.

Subgroups that were reported included individuals with problematic opioid use and a comorbid substance use condition, mental health condition, or chronic low back pain. Pregnant women, veterans, and individuals positive for HIV were also captured in studies. Very few statistically significant or substantive differences were found between intervention and control groups for specialty populations on any outcomes measured, and, unfortunately, few studies focused on specialty populations.

## Discussion

### Summary of findings

The opioid crisis remains of great concern and efforts to maximize the effectiveness of treatment for individuals with OUD are urgently needed. In this study we performed a systematic review that included 72 trials that compared psychosocial interventions among individuals receiving OAT, with the goal of conducting NMAs to establish a hierarchy of treatment strategies. Unfortunately, due to variability in outcomes assessed as well as the formats of evaluation and reporting, only one outcome (treatment retention) could be analyzed using NMA methods. Rewards-based interventions, specifically CM alone or in tandem with counselling or CRA, appeared most effective for treatment retention and were significantly more effective compared to OAT-only. SUCRA rankings for interventions were also generated, however, most psychosocial interventions were administered in a single study with few included patients. This limits the ability for robust conclusions to be drawn about the superiority of other psychosocial interventions. The co-primary outcome of interest, opioid use, including studies that reported this as opioid use or opioid abstinence as measured by urinalysis, could not be meta-analyzed given the considerable diversity in reporting formats that was encountered. The majority of included studies did not find a statistically significant benefit of adding psychosocial components to standard OAT for reducing opioid use.

As a consequence of considerable between-study variability in formats of outcome evaluation and reporting, findings from our a priori secondary outcomes of interest were primarily synthesized using a descriptive approach. The majority of outcomes we assessed, including other drug use, mental health symptoms, alcohol use, adherence to OAT, self-reported opioid use, HIV/HCV risk behavior, withdrawal symptoms, adverse events, dropouts from psychotherapy, measures of craving, and quality-of-life outcomes were associated with a lack of statistically significant differences between intervention groups. Relapse prevention, reported by just two studies, was one exception wherein an added benefit of psychosocial interventions (i.e., CBT or mindfulness-based stress reduction) was observed, in that fewer individuals relapsed compared to OAT-only.

### Findings in context

To our knowledge, this is the first systematic review of psychosocial interventions used as an adjunct to OAT to quantitatively combine available evidence for treatment retention. In 2016, a systematic review of psychosocial interventions used in conjunction with OAT included 27 studies that were qualitatively synthesized [25]. The authors of the review also identified variability in the delivery of interventions and study outcomes, and concluded that considerable gaps existed in the literature. In a narrative review that studied the role of behavioural interventions along with buprenorphine treatment, the authors described a need to enhance treatment retention given the high dropout rates found in studies [27]. The authors also described some benefit from behavioural interventions, specifically CM, and recommended its application within a stepped care model. Both aforementioned reviews supported the efficacy of providing psychosocial interventions in addition to OAT, while noting variability within studies,

and did not provide quantitative syntheses. One recent umbrella review focused on the management of OUD in a primary care setting [120]. One outcome of interest was treatment retention, whereby retention improved when counselling or contingency management was added to OAT, although the comparative effectiveness of psychosocial interventions was not tested in this review [120]. A Cochrane review of psychosocial interventions and OAT for opioid dependence was conducted in 2011 and included 28 RCTs [24]. Within, 22 studies that assessed treatment retention and were meta-analyzed in this review, no statistically significant differences were found when psychosocial interventions were incorporated into treatment. Our findings provide updated evidence upon which clinically relevant recommendations related to psychosocial interventions can be made. Importantly, our review also presents an overview of the limitations of the available evidence upon which future trials should strive to improve.

## Limitations

There are limitations of the current review that should be noted, and they relate to the study populations enrolled, RoB, the treatments compared, and the outcomes measured. First, with regard to study populations, in our efforts to seek out available data within key clinical subgroups, it became apparent that studies often excluded certain types of individuals (e.g., pregnant women, individuals with comorbid mental health concerns). Less than 10% of the included evidence base specifically recruited individuals with comorbid mental health conditions, and only one study was focused on individuals with chronic pain using prescription opioids [103]. There were also no eligible trials that aimed to study individuals who were incarcerated or youths. Many of these populations are susceptible to OUD and have been highlighted as having unique needs for treatment [121, 122]. Furthermore, the effectiveness of interventions among those with OUD has also been shown to differ by sex [123]. Within our set of included trials, study populations were predominantly composed of men, with almost a third comprising at least 75% males. Merely one study stratified all analyses by sex [77]; its findings highlighted unique results by sex and reported statistically significant benefits of psychosocial interventions used adjunctive to OAT specifically amongst women. The available evidence may not generalize well to individuals that would typically present in clinical settings [124]. Trials that include individuals with comorbid conditions are urgently needed to reliably compare psychosocial therapies in real world settings.

Second, the Cochrane RoB tool identified several limitations of the included studies. For the co-primary outcome of treatment retention, the RoB for the selective outcome reporting item was rated as unclear for 54% of studies due to a lack of trial registration which precluded the comparison of registered outcomes to published results. For 37% of studies which measured treatment retention, there was a high RoB, suggesting that selective outcome reporting occurred (e.g., changing the way that treatment retention was measured). Therefore, selective outcome reporting for this outcome cannot be ruled out and may impact the results of the NMA.

Third, regarding the interventions compared, components of psychosocial interventions (such as the community reinforcement approach, or CRA) differed between studies. For example, some CRA interventions incorporated modules for skills training (e.g., assertiveness skills, self-management skills), while others focused on engaging in non-drug related activities. Variability in frequency and duration of sessions and follow-up times were present amongst studies, as was variability in the implementation of interventions. In many included studies, details necessary for an intervention to be utilized in healthcare systems (e.g., frequency of reward provided) were absent. Inconsistencies in the implementation of interventions were present whereby the (1) content of similarly labelled interventions, (2) intensity of interventions, and (3) access to supplementary psychotherapy (e.g., group therapy), varied substantively between

studies. Next control groups were often labeled as "treatment as usual" or "usual care" and consisted of weekly counselling; however, the details of the counselling support were often not reported. Similarly, some studies reported that control groups had "access to all clinic services," but did not report the specific services that were available. This complicates comparative analyses and the interpretation of our findings, given that the intended differences between control and experimental groups were not always clear.

Next, key challenges associated with the availability of outcome data limited our ability to perform meaningful meta-analyses. There were a limited number of studies that reported many outcomes, a diversity of outcome reporting formats, and a lack of follow-up assessments. While our review included a total of 71 studies, only one outcome measure was uniformly reported in more than half of trials. Additionally, for 10 of our a priori outcome measures, fewer than 15 trials assessed and reported data of any format. Further, in the case of outcomes that were reported more frequently (i.e., between 39% and 48% of studies), such as opioid use (including opioid abstinence), illicit drug use, and abstinence from illicit drugs, inconsistency in reporting definitions (e.g., variability in what was considered abstinence from all drugs) and formats (e.g., dichotomous data and continuous data) were present more often than not. There was a lack of consistency in the outcomes reported and how they were reported across trials of OUD interventions. The follow-up for most outcomes was also relatively short, with studies measuring outcomes at 12 or 24 weeks and often immediately after the psychosocial intervention had been delivered. This limits the ability to consider the long-term effectiveness of interventions. In the context of meta-analysis and in particular NMAs, such occurrences of sparse and heterogeneous data are limiting in the types of synthesis that can be pursued.

Lastly, our work did not consider the clinical significance of findings. For instance, some studies described notable differences in outcomes, such as the proportion of urine samples that were negative for a specific drug (e.g., control group: 28% versus intervention group: 44% of urinalyses negative for cocaine), but these differences did not reach statistical significance, which may have resulted in an underestimate of the impact of psychosocial interventions as compared to control groups. Some differences found within the set of included trials may be clinically important but did not meet the threshold for statistical significance perhaps because many of the studies were of limited sample size.

Future research in this area should be designed in consideration of the aforementioned limitations of the available evidence. Future reviews may wish to include articles that are published in any language to consider whether additional eligible studies can be included. In future primary studies, enrollment criteria should be designed in consideration of ways to capture populations that increase similarities with real-world clinical practice, including individuals with chronic pain and concurrent mental health conditions. Future psychosocial studies should adhere to the template for intervention description and replication (TIDieR) reporting guideline to encourage the complete reporting of intervention details that would allow for replication of methods and facilitate integration of effective interventions into clinical practice [125]. Core outcome sets represent a necessary and valuable addition to research in this area that would increase the comparability of future clinical trials and enhance the capability for cross-study syntheses by researchers in the realm of knowledge synthesis [126].

## Conclusions and policy implications

Integrating rewards-based interventions such as CM, in addition to OAT, for OUD appears to be more efficacious than OAT-only for improving treatment retention. The current evidence, however, is associated with reporting limitations, high heterogeneity, and a potentially high RoB. Clear directions for future research among people with OUD have been identified and

include conducting robustly designed RCTs that (1) include key outcomes measured consistently between studies, (2) include individuals with comorbid psychiatric and physical disorders, and (3) are adequately reported to allow for the application of effective interventions within clinical practice. Given the urgency of the opioid crisis, clinicians and healthcare centres aiming to improve the treatment of OUD can consider implementing CM in addition to OAT to increase retention in treatment.

## Supporting information

**S1 Data.**
(XLSX)

**S1 Text. Completed PRISMA for network meta-analyses checklist.**
(DOCX)

**S2 Text. Literature search strategy.**
(DOCX)

**S3 Text. A priori list of eligible interventions.**
(DOCX)

**S4 Text. Additional details, statistical methods for NMA.**
(DOCX)

**S5 Text. Protocol deviations.**
(DOCX)

**S6 Text. Outcomes reported by study.**
(DOCX)

**S7 Text. Risk of bias evaluations.**
(DOCX)

**S8 Text. Overview of findings by study, treatment retention (not included in network meta-analyses).**
(DOCX)

**S9 Text. Overview of findings by study, abstinence from opioids.**
(DOCX)

**S10 Text. Overview of findings by study, opioid use—Urinalysis.**
(DOCX)

**S11 Text. Overview of findings by study, drug use.**
(DOCX)

**S12 Text. Overview of findings by study, self-reported illicit drug use.**
(DOCX)

**S13 Text. Overview of findings by study, abstinence from illicit drugs.**
(DOCX)

**S14 Text. Overview of findings by study, mental health.**
(DOCX)

**S15 Text. Overview of findings by study, alcohol use.**
(DOCX)

**S16 Text. Overview of findings by study, adherence to OAT.**
(DOCX)

**S17 Text. Overview of findings by study, self-reported opioid use.**
(DOCX)

**S18 Text. Overview of findings by study, HIV/HCV-risk taking behaviours.**
(DOCX)

**S19 Text. Overview of findings by study, withdrawal symptoms.**
(DOCX)

**S20 Text. Overview of Findings by study, adverse events.**
(DOCX)

**S21 Text. Overview of findings by study, dropouts from psychotherapy.**
(DOCX)

**S22 Text. Overview of findings by study, cravings.**
(DOCX)

**S23 Text. Overview of findings by study, relapse.**
(DOCX)

**S24 Text. Overview of findings by study, quality of life.**
(DOCX)

**S25 Text. Included studies.**
(DOCX)

**S26 Text. Excluded studies from full text screening (with reasons for exclusion).**
(DOCX)

**S27 Text. Treatment retention NMA—Model fit statistics and findings from sensitivity analyses.**
(DOCX)

**S28 Text. League table: Unadjusted RE NMA, treatment retention.**
(DOCX)

**S29 Text. Narrative summary of findings for secondary outcomes with data from 15 studies or fewer.**
(DOCX)

## Acknowledgments

We would like to thank Raymond Daniel (Assistant Information Specialist) for his role in providing support in database management and document procurement, and Fatemeh Yazdi and Misty Pratt for their role in study screening and data extraction. We would also like to thank Candyce Hamel, Micere Thuku, and Dr. Kelly Cobey for their assistance and input on risk of bias ratings, and Beverley Shea on providing input on this project.

**Disclaimer:** The funders had no role in the design of the planned study or preparation of this manuscript.

## Author Contributions

**Conceptualization:** Danielle Rice, Kimberly Corace, Brian Hutton.

**Data curation:** Danielle Rice, Dianna Wolfe, Leila Esmaeilisaraji, Alan Michaud, Alicia Grima, Bradley Austin, Reuben Douma, Pauline Barbeau, Claire Butler, Becky Skidmore, Brian Hutton.

**Formal analysis:** Danielle Rice, Dianna Wolfe, Brian Hutton.

**Funding acquisition:** Danielle Rice, Kimberly Corace, Melanie Willows, Patricia A. Poulin, Beth A. Sproule, Amy Porath, Gary Garber, Sheena Taha, Gord Garner, David Moher, Kednapa Thavorn, Brian Hutton.

**Methodology:** Danielle Rice, Dianna Wolfe, Becky Skidmore, Brian Hutton.

**Project administration:** Danielle Rice, Brian Hutton.

**Resources:** Kimberly Corace, Brian Hutton.

**Software:** Brian Hutton.

**Supervision:** Kimberly Corace, Brian Hutton.

**Validation:** Danielle Rice.

**Visualization:** Brian Hutton.

**Writing – original draft:** Danielle Rice, Brian Hutton.

**Writing – review & editing:** Danielle Rice, Kimberly Corace, Dianna Wolfe, Leila Esmaeilisaraji, Alan Michaud, Alicia Grima, Bradley Austin, Reuben Douma, Pauline Barbeau, Claire Butler, Melanie Willows, Patricia A. Poulin, Beth A. Sproule, Amy Porath, Gary Garber, Sheena Taha, Gord Garner, Becky Skidmore, David Moher, Kednapa Thavorn, Brian Hutton.

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
