## [Decision Letter · Decision Letter 0]

23 Oct 2020

PONE-D-20-28360

­Evaluating Comparative Effectiveness of Psychosocial Interventions Adjunctive to Opioid Agonist Therapy for Opioid Use Disorder: A Systematic Review with Network Meta-Analyses

PLOS ONE

Dear Dr. Hutton,

Thank you for submitting your manuscript to PLOS ONE. After careful consideration, we feel that it has merit but does not fully meet PLOS ONE’s publication criteria as it currently stands. Therefore, we invite you to submit a revised version of the manuscript that addresses the points raised during the review process.

We look forward to receiving your revised manuscript.

Kind regards,

Tim Mathes

Academic Editor

PLOS ONE

Journal Requirements:

2. We note that the search included publications published before January 2019. Systemeatic reviews and meta-analyses submitted to PLOS ONE should include studies published in the last 12 months. Please update your search to include studies included in this time period.

"BH has previously received honoraria from Eversana (previously Cornerstone Research Group) for the provision of methodologic advice related to systematic reviews and meta-analysis. The remaining authors have declared that no competing interests exist."

Reviewers' comments:

Reviewer's Responses to Questions

**Comments to the Author**

1. Is the manuscript technically sound, and do the data support the conclusions?

Reviewer #1: Yes

Reviewer #2: Yes

2. Has the statistical analysis been performed appropriately and rigorously? 

Reviewer #1: Yes

Reviewer #2: Yes

3. Have the authors made all data underlying the findings in their manuscript fully available?

Reviewer #1: Yes

Reviewer #2: Yes

4. Is the manuscript presented in an intelligible fashion and written in standard English?

Reviewer #1: Yes

Reviewer #2: Yes

5. Review Comments to the Author

Reviewer #1: The purpose of this study was to evaluate the comparative effectiveness of psychosocial interventions adjunctive to opioid agonist therapy for opioid use disorder. This is a topic of great interest because opioid use disorder has become a public health crisis.

The authors of the study conducted a systematic review and a network meta-analysis (NMA) of randomized controlled trials (RCTs). Overall, the manuscript is well organized and written clearly. The methods are well described and consistent with the study protocol declared a priori on the PROSPERO database. The study complies with the guidelines for conducting and reporting a systematic review and meta-analysis.

The main strength of this study is to highlight the considerable heterogeneity in the outcomes used to assess the efficacy of interventions in the clinical trials, which in most cases prevents any reliable quantitative synthesis to be made. Further to this, in many RCTs, interventions are poorly described, and there is a great need to improve quality of reporting in psychosocial studies. The proposals of the authors to improve the quality of research (e.g. TIDieR guidelines, core outcome sets) are necessary to provide policy makers reliable data on the efficacy of psychosocial interventions adjunctive to opioid agonist therapy.

I only have a few minor comments and suggestions:

1) As network meta-analyses (NMA) were unlikely to produce reliable findings for almost all outcomes, the authors performed a NMA only for the most commonly reported outcome (i.e. treatment retention). As this outcome is missing for about one third of studies, a selective outcome reporting bias cannot be ruled out. The authors should precise this point in the discussion.

2) The highest ranked treatments based on SUCRA values were interventions which were assessed in almost all cases by only one study with very small number of patients. These values do not in any way allow conclusions to be drawn about the superiority of these interventions. To avoid any misinterpretation, this should be explicitly stated in the discussion.

3) It could be interesting to discuss another limitation of randomized controlled trials included in the systematic review, namely the limited duration of patient follow-up (the majority of follow-up time points was 12 or 24 weeks, with a median = 24 weeks). This issue is also found in other areas such as the treatment of alcohol use disorders. While the management of a substance use disorder implies long-term follow-up of the patient, clinical trials often assess the efficacy of an intervention over a short period of time, which does not make it possible to determine whether the effect of the intervention is maintained in the long term.

4) Are there any studies included in the systematic review for which the opioid agonist therapy (OAT) differed between arms (e.g. one arm with OAT corresponding to methadone and another arm corresponding to buprenorphine)? If so, the differences in effect sizes could also result from differences between pharmacological treatments, and not just a difference between psychosocial interventions. Could the authors clarify this point?

5) It was initially planned that quasi-experimental studies would be included. In the end, did the authors choose to keep only randomized controlled trials?

6) In a secondary analysis, the authors performed a meta-regression to adjust for the control group risk. Could the authors clarify what they mean by control group risk?

7) Nine articles were excluded because they were not published in English or French. This may have introduced some bias in the analyses.

8) Please check the numbers of the flow chart (there is a discrepancy between the number of records screened, the number of records excluded and the number of full-text articles assessed for eligibility).

9) Please check the percentage of studies assessing opioid use through urinalysis p30l8 (the denominator is not 71).

Reviewer #2: The authors performed a network meta-analysis to assess the comparative efficacy between different psychosocial interventions added to Opioid Agonist Therapy with the aim to find the most appropriate psychosocial therapy to apply as an adjunct to OAT. The clinical question is very relevant and left unanswered by previous systematic reviews. The review is well done and very well reported. The characteristics of the included studies and the results are reported in a very detailed and comprehensive way. Only minor revisions are suggested:

Abstract

Despite the authors in the background correctly underline the importance of NMA as a statistical method that allows comparisons between several different treatments and the identification of the most efficacious, and despite the fact that they state in the abstract they do not report the most interesting result results of NMA, i.e. the results of the comparative efficacy between the different psychosocial treatments considered, but simply state that “statistically significant differences were found when psychosocial interventions were used as an adjunct to OAT as compared to OAT-only”. Then they report that “The addition of rewards-based interventions such as contingency management (alone or with community reinforcement approach) to OAT was superior to OAT-only.” No information is provided about the effectiveness/un-effectiveness of the other interventions, the comparative efficacy between treatments and the most efficacious psychosocial treatment. The results section of the abstract does not reflect the added value of a NMA and the primary objective of the review, i.e. “to compare the relative benefits and harms of psychosocial therapies among people with OUD receiving OAT”.

Background, row 16: in the sentence “Previous systematic and narrative reviews 25,27 the reference n 24 should be added, as it is a systematic review with meta-analysis assessing the efficacy of psychosocial intervention combined with maintenance treatment in comparison with maintenance treatment alone

Results. Row 27-29: the authors reported that “the most common comparator groups were counselling (n = 40/148 arms, 27.0%) and OAT-only (n = 16/148, 10.8%).” I suggest to specify “the most common comparator groups were counselling+OAT “for clarity, given that it is stated above that “all eligible study arms included OAT”. Moreover, information about comparator is reported only for the 37.8% of studies. What about the comparator of the remaining 62.2% of studies?

Table 1: I suggest to add two further columns with the description of the experimental intervention and the comparator

Table 2 and tables S8-S24: I suggest to change the name of the tables: the wording “Summary of findings” is normally used to report the results of assessment of the certainty/ quality of the evidence according to the GRADE methodology in Cochrane Reviews, so in this context it is misleading, as you did not assess certainty of evidence but simply report narratively the results for the secondary outcomes

Discussions: Limitation of the review: impact of risk of biases of the included studies on the validity of the results and conclusions is not discussed

Finding in context: the reference n 24 should be cited and discussed as well, together with the other systematic reviews that assessed the efficacy of psychosocial interventions combined with OAT. Though it is an old review published in 2011, it is a Cochrane Review.

6. PLOS authors have the option to publish the peer review history of their article (what does this mean?). If published, this will include your full peer review and any attached files.

Reviewer #1: No

Reviewer #2: No

---

## [Author Response · Author response to Decision Letter 0]

7 Dec 2020

Editor Comments

• Response: We have revised the formatting of our manuscript to align with PLOS ONE’s style requirements. 

2. We note that the search included publications published before January 2019. Systemeatic reviews and meta-analyses submitted to PLOS ONE should include studies published in the last 12 months. Please update your search to include studies included in this time period.

• Response: As requested, we have updated our search to include studies published in the last 12 months. Our search now includes articles published until the end of June 2020. One new eligible article has been included and this trial did not change the conclusions for our review.

"BH has previously received honoraria from Eversana (previously Cornerstone Research Group) for the provision of methodologic advice related to systematic reviews and meta-analysis. The remaining authors have declared that no competing interests exist."

• Response: We have revised the manuscript statement to note that there are no restrictions on the sharing of data and/or materials. The revised statement notes that the conflict of interest does not alter our adherence to PLOS ONE policies on sharing data and materials. This statement can be found in the cover letter and on page 43 of the revised manuscript. 

• Response: We have added a caption to each of the figures. 

Reviewer 1 Comments

Reviewer #1: The purpose of this study was to evaluate the comparative effectiveness of psychosocial interventions adjunctive to opioid agonist therapy for opioid use disorder. This is a topic of great interest because opioid use disorder has become a public health crisis.

The authors of the study conducted a systematic review and a network meta-analysis (NMA) of randomized controlled trials (RCTs). Overall, the manuscript is well organized and written clearly. The methods are well described and consistent with the study protocol declared a priori on the PROSPERO database. The study complies with the guidelines for conducting and reporting a systematic review and meta-analysis. The main strength of this study is to highlight the considerable heterogeneity in the outcomes used to assess the efficacy of interventions in the clinical trials, which in most cases prevents any reliable quantitative synthesis to be made. Further to this, in many RCTs, interventions are poorly described, and there is a great need to improve quality of reporting in psychosocial studies. The proposals of the authors to improve the quality of research (e.g. TIDieR guidelines, core outcome sets) are necessary to provide policy makers reliable data on the efficacy of psychosocial interventions adjunctive to opioid agonist therapy.

• Response: We thank the reviewer for their positive review of our manuscript.

I only have a few minor comments and suggestions:

1) As network meta-analyses (NMA) were unlikely to produce reliable findings for almost all outcomes, the authors performed a NMA only for the most commonly reported outcome (i.e. treatment retention). As this outcome is missing for about one third of studies, a selective outcome reporting bias cannot be ruled out. The authors should precise this point in the discussion.

• Response: We have added a paragraph within the limitations section about the potential selective outcome reporting that took place the NMA. This paragraph reads: “Second, the Cochrane RoB tool identified several limitations of the included studies. For the co-primary outcome of treatment retention, the RoB for the selective outcome reporting item was rated as unclear for 54% of studies due to a lack of trial registration which precluded the comparison of registered outcomes to published results. For 37% of studies which measured treatment retention, there was a high RoB, suggesting that selective outcome reporting occurred (e.g., changing the way that treatment retention was measured). Therefore, selective outcome reporting for this outcome cannot be ruled out and may impact the results of the NMA.”

2) The highest ranked treatments based on SUCRA values were interventions which were assessed in almost all cases by only one study with very small number of patients. These values do not in any way allow conclusions to be drawn about the superiority of these interventions. To avoid any misinterpretation, this should be explicitly stated in the discussion.

• Response: We thank the reviewer for this suggestion. We have added an explicit statement about this within the first paragraph of the discussion that describes the findings. The statement reads, “[…] Rewards-based interventions, specifically CM alone or in tandem with counselling or CRA, appeared most effective for treatment retention and were significantly more effective compared to OAT-only. SUCRA rankings for interventions were also generated, however, most psychosocial interventions were administered in a single study with few included patients. This limits the ability for robust conclusions to be drawn about the superiority of other psychosocial interventions.”

3) It could be interesting to discuss another limitation of randomized controlled trials included in the systematic review, namely the limited duration of patient follow-up (the majority of follow-up time points was 12 or 24 weeks, with a median = 24 weeks). This issue is also found in other areas such as the treatment of alcohol use disorders. While the management of a substance use disorder implies long-term follow-up of the patient, clinical trials often assess the efficacy of an intervention over a short period of time, which does not make it possible to determine whether the effect of the intervention is maintained in the long term.

• Response: We have incorporated this limitation into the discussion section when discussing the limits of the outcomes collected. We have added a statement that reads, “The follow-up for most outcomes was also relatively short, with studies measuring outcomes at 12 or 24 weeks and often immediately after the psychosocial intervention had been delivered. This limits the ability to consider the long-term effectiveness of interventions.”

4) Are there any studies included in the systematic review for which the opioid agonist therapy (OAT) differed between arms (e.g. one arm with OAT corresponding to methadone and another arm corresponding to buprenorphine)? If so, the differences in effect sizes could also result from differences between pharmacological treatments, and not just a difference between psychosocial interventions. Could the authors clarify this point?

• Response: We did not include any studies that had different types of OAT delivered due to this concern. Specifically, if a study had one arm with methadone (+ psychosocial intervention) and the other arm with buprenorphine (+ psychosocial intervention) it was excluded. We have revised the wording and added additional clarification to the methods section to try and clearly articulate this point. Please see page 8, lines 4-6 of the tracked changes version which reads, “Studies that did not include at least two arms receiving the same pharmacological interventions were excluded, given best practice guidelines which include OAT as first line treatment for OUD.” And please see page 8, lines 17-18 of the tracked changes version which reads, “If studies included more than two groups with different pharmacological interventions (e.g., two groups randomized to methadone and psychosocial interventions and two groups with buprenorphine and psychosocial interventions), we included only two study arms that applied the same pharmacological intervention based on the OAT that was most frequently reported across all studies.”

5) It was initially planned that quasi-experimental studies would be included. In the end, did the authors choose to keep only randomized controlled trials?

• Response: We had initially planned to include quasi-experimental studies in case there was a lack of RCTs identified. Given the volume of eligible RCTs that were identified, we were able to restrict our sample to only RCTs to reduce the potential bias from including quasi-experimental studies. This decision was also made to enhance the feasibility of this project which was funded as a rapid 6-month review grant. 

6) In a secondary analysis, the authors performed a meta-regression to adjust for the control group risk. Could the authors clarify what they mean by control group risk?

• Response: We’re happy to explain further. In a network of studies with a predominant control group that links most of the interventions together (for example, a network of placebo-controlled studies where few or no trials directly comparing the active interventions are available), variability in the placebo response rates across studies can be investigated as a means of identifying signs of differences in the different study populations; the premise is that rates of outcomes (for example, mortality) should be comparable across studies/treatment comparisons if the populations within the studies are truly comparable. Fitting a network meta-regression analysis accounting for these event rates in the linking control group allows the analyst to investigate whether accounting for this quantity changes findings in an important way; in cases where adding this adjustment into the NMA model does not result in material changes to the estimated treatment effects, then concerns over important differences between study populations and any effect on the findings are unwarranted. To supplement the above description, we have added the following text to the methods section of the manuscript in parentheses where we make mention of the control group risk adjustment: “as a proxy to consider between-study differences in multiple confounders.”

7) Nine articles were excluded because they were not published in English or French. This may have introduced some bias in the analyses.

• Response: We agree that this could introduce bias. While it is unknown if these articles would have met our inclusion criteria, we have added a recommendation that future reviews include studies available in any language. This statement reads, “Future reviews may wish to include articles that are published in any language to consider whether additional eligible studies can be included.”

8) Please check the numbers of the flow chart (there is a discrepancy between the number of records screened, the number of records excluded and the number of full-text articles assessed for eligibility).

• Response: We have revised the numbers of the flow chart to represent the correct values, including the updated search numbers.

9) Please check the percentage of studies assessing opioid use through urinalysis p30l8 (the denominator is not 71).

• Response: We thank the reviewer for noting this, we have revised the percentage reported for opioid use through urinalysis. 

Reviewer 2 Comments

Reviewer #2: The authors performed a network meta-analysis to assess the comparative efficacy between different psychosocial interventions added to Opioid Agonist Therapy with the aim to find the most appropriate psychosocial therapy to apply as an adjunct to OAT. The clinical question is very relevant and left unanswered by previous systematic reviews. The review is well done and very well reported. The characteristics of the included studies and the results are reported in a very detailed and comprehensive way. Only minor revisions are suggested:

• Response: We thank the reviewer for their positive review of our manuscript.

Abstract

Despite the authors in the background correctly underline the importance of NMA as a statistical method that allows comparisons between several different treatments and the identification of the most efficacious, and despite the fact that they state in the abstract they do not report the most interesting result results of NMA, i.e. the results of the comparative efficacy between the different psychosocial treatments considered, but simply state that “statistically significant differences were found when psychosocial interventions were used as an adjunct to OAT as compared to OAT-only”. Then they report that “The addition of rewards-based interventions such as contingency management (alone or with community reinforcement approach) to OAT was superior to OAT-only.” No information is provided about the effectiveness/un-effectiveness of the other interventions, the comparative efficacy between treatments and the most efficacious psychosocial treatment. The results section of the abstract does not reflect the added value of a NMA and the primary objective of the review, i.e. “to compare the relative benefits and harms of psychosocial therapies among people with OUD receiving OAT”.

• Response: We agree that providing additional information in the abstract would be ideal. We are, however, already ~100 words over PLOS ONE’s recommended limit for the abstract word count. Further, as highlighted by reviewer 1, other comparisons were assessed in often only one study with very small numbers of patients. We do not want to draw conclusions about the superiority of interventions where these could be misinterpreted. After presenting the results of psychosocial interventions as compared to OAT-only, we have added a statement in the abstract results section that reads, “Few statistically significant differences were identified among any other pairwise comparisons.” We hope that this will provide readers with additional information from the abstract without overstating our findings.

Background, row 16: in the sentence “Previous systematic and narrative reviews 25,27 the reference n 24 should be added, as it is a systematic review with meta-analysis assessing the efficacy of psychosocial intervention combined with maintenance treatment in comparison with maintenance treatment alone

• Response: We have added reference 24 to the statement within the introduction. 

Results. Row 27-29: the authors reported that “the most common comparator groups were counselling (n = 40/148 arms, 27.0%) and OAT-only (n = 16/148, 10.8%).” I suggest to specify “the most common comparator groups were counselling+OAT “for clarity, given that it is stated above that “all eligible study arms included OAT”. Moreover, information about comparator is reported only for the 37.8% of studies. What about the comparator of the remaining 62.2% of studies?

• Response: We thank the reviewer for this recommendation, we have made a number of revisions to try and address this comment. First, we have added that the control group, counselling, was in addition to OAT. Next, we appreciate the request for more information about the control groups. We have revised the control group statement to only consider one control group for each of the included studies (N = 72, based on updated search). The denominator now more clearly represents how many control groups there are, as studies have multiple intervention arms, but can be compared against one main control group. As such, we have clarified the statement the say, “while the most common control groups were counselling in addition to OAT (n = 44/72 control groups, 59.5%)], OAT-only (n = 17/72 control groups, 23.0%)”. This now accounts for 82.5% of control groups, the remaining control groups were education, counselling+education, counselling+contingency management, cognitive behavioural therapy (alone or with counselling), and counselling+12 step facilitation therapy.

Table 1: I suggest to add two further columns with the description of the experimental intervention and the comparator.

• Response: We thank the reviewer for this recommendation and agree that providing readers with this information on a study by study basis is helpful. As there are many studies that included multiple arms and components of the intervention we have added a statement to direct readers to where the comprehensive information can be found which includes all group intervention types including the type of psychosocial intervention, OAT, and the description of each intervention. In the results section we have added the following statement to the results section, “Table 1 provides a study-by-study account of additional information including population and key demographics and S1 data file provides detailed accounts of the study accounts of the intervention and comparator groups.” 

Table 2 and tables S8-S24: I suggest to change the name of the tables: the wording “Summary of findings” is normally used to report the results of assessment of the certainty/ quality of the evidence according to the GRADE methodology in Cochrane Reviews, so in this context it is misleading, as you did not assess certainty of evidence but simply report narratively the results for the secondary outcomes

• Response: We appreciate the reviewers concern over wording here. We have changed the titles of all tables to be worded instead as “Overview of findings.”

Discussions: Limitation of the review: impact of risk of biases of the included studies on the validity of the results and conclusions is not discussed

• Response: Reviewer 1 also recommended that we note limitations from the RoB tool. We have added a paragraph within the limitations section about the impact of RoB of included studies, and have also highlighted the the potential selective outcome reporting that took place the NMA. This paragraph reads: “Second, the Cochrane RoB tool identified several limitations of the included studies. For the co-primary outcome of treatment retention, the RoB for the selective outcome reporting item was rated as unclear for 54% of studies due to a lack of trial registration which precluded the comparison of registered outcomes to published results. For 37% of studies which measured treatment retention, there was a high RoB, suggesting that selective outcome reporting occurred (e.g., changing the way that treatment retention was measured). Therefore, selective outcome reporting for this outcome cannot be ruled out and may impact the results of the NMA.”

Finding in context: the reference n 24 should be cited and discussed as well, together with the other systematic reviews that assessed the efficacy of psychosocial interventions combined with OAT. Though it is an old review published in 2011, it is a Cochrane Review.

• Response: As recommended, we have added the 2011Cochrane review findings into the discussion section of the review. The statement reads, “A cochrane review of psychosocial interventions and OAT for opioid dependence was conducted in 2011 and included 28 RCTs.[24] Within 22 studies that assessed treatment retention and were meta-analyzed in this review, no statistically significant differences were found when psychosocial interventions were incorporated into treatment. Our findings provide updated evidence upon which clinically relevant recommendations related to psychosocial interventions can be made.”

---

## [Editor Report · Decision Letter 1]

9 Dec 2020

­Evaluating Comparative Effectiveness of Psychosocial Interventions Adjunctive to Opioid Agonist Therapy for Opioid Use Disorder: A Systematic Review with Network Meta-Analyses

PONE-D-20-28360R1

Dear Dr. Hutton,

We’re pleased to inform you that your manuscript has been judged scientifically suitable for publication and will be formally accepted for publication once it meets all outstanding technical requirements.

Kind regards,

Tim Mathes

Academic Editor

PLOS ONE
---

## [Editor Report · Acceptance letter]

14 Dec 2020

PONE-D-20-28360R1 

­Evaluating comparative effectiveness of psychosocial interventions adjunctive to opioid agonist therapy for opioid use disorder: A systematic review with network meta-analyses 

Dear Dr. Hutton:

I'm pleased to inform you that your manuscript has been deemed suitable for publication in PLOS ONE. Congratulations! Your manuscript is now with our production department. 

Kind regards, 

on behalf of

Dr. Tim Mathes 

Academic Editor

PLOS ONE